- Three-Dimensional Biomass Burning Emission Inventory for
- Southeast and East Asia Based on Multi-Source Data Fusion and
- Machine Learning
- Yinbao Jin<sup>1,5</sup>, Heng Huang<sup>2\*</sup>, Jian Liu<sup>3,4</sup>, Yiming Liu<sup>5</sup>, Xiaoyang Chen<sup>6</sup>, Yongqiang, Chen<sup>1</sup>, Licheng Li<sup>1</sup>,
- Oi Fan<sup>5,7,8,\*</sup>

- Guangzhou Meteorological Satellite Ground Station (Guangdong Meteorological Satellite Remote
- Sensing Center), Guangzhou, 510640, China
- 2School of Geomatics, Liaoning Technical University, Fuxin 123000, China
- <sup>3</sup>College of Environment and Ecology, Taiyuan University of Technology, Taiyuan 030024, China
- <sup>4</sup>Shanxi Key Laboratory of Complex Air Pollution Control and Carbon Reduction, Taiyuan University
- of Technology, Taiyuan 030024, China
- School of Atmospheric Sciences, Sun Yat-Sen University, Zhuhai 519082, China
- 6Institute of Tropical and Marine Meteorology, China Meteorological Administration, Guangzhou
- 510000, China
- Southern Marine Science and Engineering Guangdong Laboratory, Zhuhai 519082, China
- Suangdong Province Key Laboratory for Climate Change and Natural Disaster Studies, School of
- Atmospheric Sciences, Sun Yat-sen University, Guangzhou 510275, China

29

- $20 \qquad *\textit{Correspondence to} : Qi \ Fan \ (eesfq@mail.sysu.edu.cn) \ and \ Heng \ Huang \ (huangheng@lntu.edu.cn)$
- Abstract
- Biomass burning (BB) is a major source of atmospheric pollutants in Southeast and East Asia (SEA), yet
- most existing emission inventories lack accurate diurnal cycles and vertical injection profiles, limiting
- the accuracy of air quality and climate simulations. This study develops the Southeast and East Asia Fire
- (SEAF) inventory, an hourly 3 km three-dimensional (3D) emission dataset for 2023, by fusing fire
- radiative power (FRP) from Himawari-8/9 AHI and VIIRS through cloud correction, cross-calibration,
- and a region-vegetation-specific Gaussian diurnal reconstruction with dynamic gap filling. Vertical
- profiles are further constrained using a random forest (RF) Shapley Additive Explanations (SHAP)

framework trained with Multi-angle Imaging SpectroRadiometer (MISR) smoke plume heights (SPH)

- and ERA5 meteorology. The SEAF inventory exhibited strong consistency with TROPOMI CO, showing
- a correlation of R = 0.97 in monthly columns and differing by only 7.81% during a representative event
- on 9 March 2023. Annual PM<sub>2.5</sub> emissions in SEAF are approximately 2362 Gg y<sup>-1</sup>, which is 67% lower
- than the Fire INventory from NCAR (FINN) but aligns well with the Fire Energetics and Emissions

- Research (FEER) and the Quick Fire Emissions Dataset (QFED) estimates. The RF-SHAP framework
- successfully predicted SPH, with over 90% of estimates within ± 500 m. This approach corrects the near-
- surface overweighting of conventional schemes by reducing emissions below 0.3 km and enhancing
- injection between 2.7-5.5 km during the spring burning peak, yielding vertical profiles that closely align
- with satellite observations. SHAP analysis identified temperature- and radiation-related factors, 38
- particularly the vertical integral of temperature (Vit) and terrain elevation, as the primary drivers of SPH,
- with additional contributions from FRP, planetary boundary layer height, and seasonal-meteorological 40
- interactions. These advances in both diurnal timing and vertical injection are anticipated to provide an
- observation-driven, hourly 3D BB emission dataset for SEA that can improve the reliability of air quality,
- climate, and policy assessment models.

#### 1. Introduction

60

68

- Open biomass burning (BB) exerts substantial impacts on climate, ecosystems, economies, and public
- health by releasing large quantities of aerosols and trace gases into the atmosphere (Pullabhotla et al.,
- 2023; Reining et al., 2025; Yu et al., 2022). Black carbon (BC) and primary organic aerosols (POA)
- account for approximately 40-59% and 60-85% of BB aerosol emissions, respectively, while non-
- methane organic gases and greenhouse gases such as methane (CH<sub>4</sub>), carbon dioxide (CO<sub>2</sub>), and nitrous
- oxide (N2O) contribute significantly to atmospheric chemistry and radiative forcing (Gkatzelis et al.,
- 2024). Through direct and indirect effects, these emissions perturb the Earth's radiative balance, alter
- aerosol composition, and modulate cloud and precipitation processes (Crutzen et al., 1979). Under global 52
- warming, wildfire frequency and intensity have increased (Brown et al., 2023; Reining et al., 2025).
- Therefore, accurate quantification of these emissions is essential for evaluating their effects on air quality,
- climate systems, guiding targeted mitigation, and improving carbon inventories (Liu and Popescu, 2022).
- Although total BB emissions in Southeast and East Asia (SEA) are lower than in southern Africa, the
- largest global source region, the regional impacts on cloud properties and radiation remain substantial
- and environmentally consequential (Ding et al., 2021). Most BB inventories in SEA are based on daily
- or monthly FRP products from polar-orbiting sensors such as the Moderate Resolution Imaging

Spectroradiometer (MODIS) and Visible Infrared Imaging Radiometer Suite (VIIRS), which miss

- pronounced diurnal cycles and nocturnal burning. These limitations bias emission estimates in regions
- with frequent cloud cover and small-scale fires (Andela et al., 2015; Liu et al., 2024b, a). Multi-source
- data fusion of geostationary and polar-orbiting FRP based on a top-down approach has emerged as an
- effective way to address these gaps (Li et al., 2022; Nguyen et al., 2023). This multi-source data fusion
- addresses the shortcomings of single data sources, providing a more accurate representation of BB
- emissions. It is particularly beneficial for regions with frequent fires or complex meteorological
- conditions, offering high-resolution data to support regional air quality simulations and significantly
- 2019). For the contiguous United States, Li et al., (2022) integrated Geostationary Operational 69

enhancing the ability to dynamically respond to variations in fire intensity and distribution (Li et al.,

- Environmental Satellite - R Series (GOES-R) and VIIRS produced an hourly 3 km inventory that better
- captured seasonal and diurnal patterns when evaluated against carbon monoxide and fine particulate
- matter from multiple inventories In East Asia, combinations of MODIS, VIIRS, and Himawari have
- yielded hourly products at kilometer scale that improve spatiotemporal characterization (Xu et al., 2023,
- and Xu et al., 2022). However, compared to other inventories, these integrated inventories still exhibit
- significant uncertainties in annual emissions (2-7 times) and monthly emissions (1-48 times) (Xu et al.,

90

91

104

105

110

reconstruct the diurnal cycle of fire emissions, which tends to understate peak burning when observations 78 are missing because of clouds or sampling gaps and may overfill periods with little activity (Li et al., 79 2019). Importantly, our previous work showed that the large spread among available inventories over 80 Southeast Asia strongly influences simulations of aerosol optical properties and radiative forcing (Jin et 81 al., 2024). These results underscore the need for improved inventories with higher spatiotemporal fidelity. 82 Accurate representation of smoke vertical injection is equally critical. Smoke plume height (SPH) 83 determines the vertical distribution of pollutants, affecting transport, removal, and radiative effects (Jin 84 et al., 2024; Li et al., 2023). Although a few models have attempted to account for SPH, they usually rely 85 on fire size and heat flux estimation, which suffers from high computational cost and bias (Driscoll et al., 86 2024). Additionally, inventories such as Integrated System for Fire Information Retrieval and Evaluation 87 (IS4FIRE) estimate SPH using semi-empirical formulas, but their accuracy is limited due to inadequate 88 consideration of meteorological variables (Driscoll et al., 2024; Freitas et al., 2007; Sofiev et al., 2012). 89 To address these challenges, researchers have gradually introduced machine learning methods to capture

the nonlinear relationships between meteorological factors and BB characteristics (Brown et al., 2023;

Wang and Wang, 2020). However, there is still a relative lack of quantitative studies on the relationship

2023b; Zhang et al., 2014). In addition, most inventories apply a climatological Gaussian scheme to

92 between SPH and meteorological factors. In-depth exploration of the driving mechanism of plume rises 93 and its integration into the three-dimensional (3D) BB emission inventory is important for improving the 94 accuracy of the emission inventory as well as the ability of global/regional atmospheric modeling. 95 This study develops the Southeast and East Asia Fire (SEAF) emission inventory, an hourly 3-km 96 resolution BB emission product covering SEA in 2023. SEAF was generated by fusing FRP data from 97 Advanced Himawari-9 Imager (AHI), VIIRS, and NOAA's Joint Polar Satellite System (NOAA20), with 98 dynamic adjustment of Gaussian-fitted FRP daily cycles to improve temporal accuracy. The inventory 99 was validated against Tropospheric Monitoring Instrument (TROPOMI) observations and five 100 commonly used BB emission inventories. A 3D version is further constructed by combining Multi-angle 101 Imaging Spectroradiometer (MISR) plume heights with ECMWF Reanalysis v5 (ERA5) meteorology 102 through a machine-learning framework. The resulting products will provide improved data support for 103 regional air-quality and climate modeling applications.

# 2. Data and Methodology

# 2.1 Satellite data

To characterize BB emissions across SEA, four representative subregions were delineated using landcover and climate classifications as shown in Figure 1. This spatial framework supports the integration of geostationary and polar-orbiting satellite observations, laying the foundation for subsequent FRPbased emission inventory development.

## 2.1.1 Satellite-based Fire Radiative Power

Satellite observations provide critical information on FRP, which serves as BB emission estimates. In this study, we mainly use two FRP products derived from polar-orbiting and geostationary satellites. The 375m I band Level 2 active fire product from the VIIRS sensor provides observations from NASA/NOAA's Suomi National Polar-orbiting Partnership (S-NPP) and NOAA-20 satellites (Schroeder et al., 2024). The VIIRS sensor was first deployed on the Suomi NPP satellite in October 2011 and has

- since been extended with support for NOAA-20 and NOAA-21 satellites (Giglio et al., 2016). The first
- fire hotspot detection from the VIIRS sensor occurred on January 19, 2012. The 375 m data supplements
- the MODIS fire detection. Both MODIS and VIIRS products show good consistency in hotspot detection,
- but the VIIRS 375 m product improves spatial resolution, offering better response for relatively smaller
- fire areas and providing improved large fire perimeter mapping(Schroeder et al., 2014). The 375m data
- product also improves nighttime performance, making these data ideal for use in support of fire
- management (e.g., near-real-time alarm systems) and other scientific applications that require improved
- fire mapping fidelity (Csiszar et al., 2014).
- The FRP data provided by the Himawari-8/9 satellite comes from its Advanced Himawari Imager (AHI)
- sensor, which has a spatial resolution of 2 km and a temporal resolution of 10 minutes, offering full disk
- coverage (60°N-60°S, 80°E-160°W) (Bessho et al., 2016). The FRP is determined using a dual-spectral
- method applied to data from the 2.3 µm and 3.9 µm bands (JAXA/EORC, 2020). Since October 2015, the
- Himawari satellite has consistently supported fire monitoring and disaster response with its high-
- precision imaging capabilities. Himawari-8 has moderate spatial resolution, very high temporal
- resolution, and a fixed observation area, making it particularly suitable for real-time monitoring of
- wildfires in SEA (Xu et al., 2022). Furthermore, unlike MODIS or Fengyun-3D fire products, which fail
- to capture nighttime events, Himawari-8 has the advantage of continuously monitoring wildfires after
- sunset, making it a valuable tool for replacing manual inspections of nighttime wildfires (Chen et al.,
- 2023). Himawari-8 is highly resistant to smoke and thin clouds, and it is very sensitive to small fires,
- providing valuable real-time fire information for wildfire management (Xu and Zhong, 2017). However,
- existing Himawari-8 fire products show poor consistency with MODIS data (Jang et al., 2019). Therefore,
- we first correct the Himawari-8/9 fire products using the methodology of Li et al., (2022) before
- performing data fusion.

# 2.1.2 TROPOMI CO

- The Tropospheric Monitoring Instrument (TROPOMI) aboard the Sentinel-5 (S5-P) satellite measures
- the total column concentration of CO, providing daily global coverage with a high spatial resolution of
- $5.5 \times 7$  km<sup>2</sup> (Landgraf et al., 2018). The instrument was launched on October 13, 2017 and can measure
- within the visible (270-500 nm), near-infrared (675-775 nm), and short-wave infrared (SWIR, 2305-2385
- 144 nm) ranges (Borsdorff et al., 2023). The TROPOMI CO data clearly shows strong sources such as
- wildfires, with a small mean difference  $(3.2 \pm 5.5\%)$  and a high correlation (R=0.97) between TROPOMI
- and CAMS (Borsdorff et al., 2018). Li et al., (2022) and Griffin et al., (2024) have validated TROPOMI
- CO as a reliable dataset for assessing the CO reliability of inverted BB emissions. In this paper, we also
- used these data and Eq. (1) to evaluate our fused inverted CO data for BB emissions.

$$M'_{co} = \sum_{i=1}^{n} \left( \rho_{sm}^{i} - \rho_{bg} \right) \times A^{i} \times M \tag{1}$$

- $M'_{co}$  represents the total CO emissions for a given fire sample.  $\rho^i_{sm}$  and  $A^i$  are the observed total column
- CO concentration (mol m<sup>-2</sup>) and the pixel area (m<sup>2</sup>) for the i smoke pixel.  $\rho_{bg}$  is the mean column density
- of background pixels. M is the molecular mass of CO ( $M = 28.01 \text{ g mol}^{-1}$ ).

166

185

## 2.1.3 MISR plume height dataset

SPH observations were obtained from the MISR Plume Height Project II, which assembled a dataset of 154 smoke plumes for the summers of 2008 to 2011 and for 2017 and 2018 (Nastan et al., 2022). The MISR 155 instrument onboard Terra acquires imagery at nine viewing angles, and heights are retrieved by stereo parallax using the MISR Interactive Explorer (MINX). Because digitization in MINX is labor intensive, 156 157 data collection was conducted over multiple years by teams at the Jet Propulsion Laboratory and the 158 Goddard Space Flight Center together with student groups at the University of Sheffield (Val Martin et al., 2018). Under favorable conditions the vertical accuracy can reach approximately 200 m (Nelson et 159 160 al., 2013). The MISR smoke plume records have been widely used to characterize wildfire injection heights and have become an important observational constraint in regional and global studies (Ke et al., 161 162 2021; Zhu et al., 2018). For this study the MISR Enhanced Research and Lookup Interface (MERLIN) 163 was used to extract 2127 plume height samples from Southeast and East Asia during 2017 and 2018 as 164 shown in Figure S1. These samples served as training data for the machine learning based estimation of 165 SPH.

### 2.2 ERA5 and Other Biomass Burning emissions

ERA5 is the fifth generation of global climate reanalysis dataset released by the European Center for 168 Medium-Range Weather Forecasts (ECMWF), which is widely used in global climate and weather 169 research. The ERA5 hourly meteorological data used in this study are sourced from the Climate Data 170 Store, offering a spatial resolution of 0.25° × 0.25°. This dataset covers several key meteorological 171 variables relevant to BB (Bell et al., 2021). Variables relevant to BB include 2m temperature, 2m dew 172 point temperature, 10m wind speed components, and precipitation, among others, which jointly provide 173 the meteorological context for fire occurrence and plume development (Dong et al., 2021; Kim et al., 174 2025; Vitolo et al., 2020). ERA5 variables were combined with FRP from VIIRS and Himawari-8/9, 175 together with MISR SPH, to train a machine learning model to predict plume injection height in 2023. 176 The set of ERA5 predictors used is summarized in Table S1. To assess the accuracy of the fused BB 177 emission inventory, a detailed spatiotemporal comparison was performed against major international 178 inventories including the Global Fire Assimilation System (GFAS), the Fire INventory from NCAR 179 (FINN), the Fire Energetics and Emissions Research (FEER), the Quick Fire Emissions Dataset (QFED), 180 and IS4FIRES (Table S2).

# 2.3 Methodologies

Figure 2 illustrates the framework for estimating hourly 3D BB emissions. Active fire observations from Himawari 9 AHI, Suomi NPP VIIRS I band, and NOAA 20 VIIRS I band are gridded at 3 km, corrected for cloud effects, and fused. Historical AHI records from 2016 to 2023 were analyzed to derive statistics on burn duration and observational gaps, which were then applied with Gaussian fitting to construct hourly FRP diurnal cycles. During intense burning periods, the diurnal curves were dynamically adjusted to recover missing peaks while preventing overestimation. Two products were generated: a 2D inventory providing surface emissions, and a 3D inventory that incorporates vertical injection by applying a RF model using FRP, ERA5 meteorology, and MISR SPH to predict injection height and allocate emissions vertically. This fused product is referred to as the Southeast and East Asia Fire (SEAF) inventory. This study focuses on 2023, a year that offered a unique convergence of consistent observations from the newly operational Himawari-9 and a scientifically significant, intense fire season driven by El Niño,

- ideal for validating the framework (Jong, 2024). Finally, the 2D SEAF inventory was evaluated against
- TROPOMI CO and six major global emission inventories, while the 3D SEAF inventory was assessed
- using MISR, Cloud-Aerosol Lidar and Infrared Pathfinder Satellite Observations (CALIPSO),
- IS4FIRES, and GFAS data.

#### 2.3.1 Data calibration

To facilitate subsequent air-quality modeling applications, all datasets were regridded to a spatial resolution of  $0.03^{\circ} \times 0.03^{\circ}$ . During aggregation, the FRP within each grid cell was summed, while the geolocation of individual fire detections was retained to represent the combined intensity of co-occurring events. This procedure is consistent with the native generation of the VIIRS I band FRP product and aligns with the objectives of emission inventory construction. For cloud correction, the cloud fraction in each grid cell at the satellite overpass time was computed using the VIIRS-I SDR terrain-corrected geolocation (GITCO) files. Cloud fraction was defined as the ratio of cloud pixels to the total number of pixels in the cell, with cloud pixels identified from their latitude and longitude coordinates. The resulting calibration was then applied to the polar-orbiting observations as specified in Eq. (2).

$$FRP^{V} = \begin{cases} \frac{FRP^{V}_{aggregation}}{1 - \beta + \alpha \times \beta^{2}} & \beta \le 95\% \\ FRP^{V}_{aggregation} & \beta > 95\% \end{cases}$$
 (2)

- where  $FRP_{aggregation}^{V}$  represents the aggregated FRP values at the grid points after regridding the
- VIIRS data.  $FRP^V$  is the cloud-corrected FRP value.  $\beta$  is the cloud fraction, and when the cloud fraction
- exceeds 95%, cloud correction is not applied to avoid overestimation.  $\alpha$  is the secondary coefficient,
- which is set to 0.25 according to the testing described in Li et al., (2022).

### 2.3.2 Geostationary satellite FRP calibration

- Given the relatively large zenith angle of the Himawari-8/9 satellite over Northeast China (Region 4,
- Figure 1), it is essential to apply appropriate calibration to the FRP data. In this study, cloud-corrected
- polar-orbiting satellite data are used as the calibration reference for the Himawari satellite at matching
- 215 times and location. The calibration formula is given by Eq. (3).

$$\overline{FRP_i^{AHI}} = FRP_i^{AHI} \times (1 + r_i) \tag{3}$$

- where  $FRP_i^{AHI}$  and  $\overline{FRP_i^{AHI}}$  represent the FRP of fire pixel i before and after calibration, respectively,
- and  $r_i$  is the calibration factor. The calibration factor is calculated based on common point pairs, which
- are obtained by matching AHI and VIIRS observations. Specifically, common point pairs are defined
- here as grid points at the same location that are simultaneously detected as fire points in both the
- geostationary and polar-orbiting fire product. "Simultaneously" indicates that the observation times differ
- by no more than  $\pm 5$  minutes. If there are multiple VIIRS detections within  $\pm 5$  minutes that match an AHI
- fire point, the point with the smallest time difference is selected for matching. For the i common point
- pair, the calibration factor is given by Eq. (4).

$$r_i = \frac{FRP_i^{VIIRS} - FRP_i^{AHI}}{FRP_i^{AHI}} \tag{4}$$

- Where  $FRP_i^{VIIRS}$  and  $FRP_i^{AHI}$  are the fire radiated power of the VIIRS and AHI of the corresponding
- point pair, respectively.
- In addition, when no common point pair is available, a dynamic calibration factor and an alternative
- calibration factor are established, depending on whether common point pairs exist at other times of the
- day for that fire pixel. Among these, the dynamic calibration factor is defined in Eq. (5)

$$r_{d,j} = \frac{1}{n_d} \sum_{i=1}^{n_d} r_i \tag{5}$$

- Where  $r_{d,j}$  is the dynamic calibration factor for pixel j on day d, and  $n_d$  is the number of common
- point pairs for that pixel on day d across different time instances.
- For fire points captured by the geostationary satellite, a significant fraction lack corresponding common
- point pairs in the polar-orbiting satellite record on the same day. To address this, an alternative calibration
- factor is introduced. Previous studies have shown that BB fuel characteristics are similar within the same
- land cover type, and that monthly climate conditions are also comparable in SEA (Huang et al., 2024;
- Yin, 2020). Therefore, the calibration factor is calculated for each month and vegetation type based on
- the available common point pairs and then averaged, as represented by Eq. (6).

$$r_{m,l} = \frac{1}{n_{m,l}} \sum_{j=1}^{n_{m,l}} r_j \tag{6}$$

- Where  $r_{m,l}$  is the alternative calibration factor for month m and land cover type l, and  $n_{m,l}$  is the
- number of common point pairs for month m and land cover type l. The results of the alternative
- calibration factors are shown in Table S3.
- Previous studies have shown that calibrated geostationary and polar-orbiting satellite data exhibit
- improved consistency, and that fusion of these datasets can compensate for limitations in the spatial and
- temporal resolution of individual datasets (Li et al., 2022; Zhang et al., 2012). For air quality modeling
- applications, the temporal resolution of the fusion was set to 1 hour, which satisfies the need for high
- temporal detail while avoiding potential incompatibility from excessively high resolution (López-Noreña
- et al., 2022). Specifically, the results were averaged within 1-hour intervals, while the spatial resolution
- was retained at 0.03°. Given the relatively high quality of polar-orbiting satellite data, these observations
- were prioritized in the fusion process, while geostationary satellite data were used to supplement missing
- detections. The specific fusion method is set in Eq. (7).

$$FRP_i^{fuse} = \begin{cases} FRP_i^{VIIRS} & VIIRS FRP > 0\\ \overline{FRP_i^{AHI}} & VIIRS FRP < 0 \text{ and } AHI FRP > 0\\ 0 & VIIRS FRP 

#### 282 2.3.4 Construction of emission inventories

- 283 The hourly FRP product, after fusion and filling, is further used to construct the Fire Radiative Energy
- (FRE) using the following Eq. (9).

$$FRE_{h,(i,j)} = \int_{t_1}^{t_2} FRP_r dt \tag{9}$$

- Where  $FRE_{n,(i,j)}$  (MJ) represents the FRE produced by the fire point (i,j) from time  $t_1$  to  $t_2$ ,  $FRP_r$
- is the reconstructed hourly FRP.
- Dry Matter (DM) refers to the weight of the material in BB that does not include water content. The
- consumption of DM is proportional to the generated flame heat (Koster et al., 2015). Based on the value
- of FRE, the DM consumed during BB over a given period can be estimated, as shown in Eq. (10).

$$DM_{h,(i,j)} = FRE_{h,(i,j)} \times F_{BC} \tag{10}$$

- Where  $DM_{h,(i,j)}$  (kg) represents the DM consumed by the fire point (i,j) during one hour of
- combustion, which is proportional to the FRE generated.  $F_{BC}$  is the biomass combustion factor. previous
- studies have shown that the relationship between DM mass of manzanita and FRE can be expressed as a
- slope of  $0.368 \pm 0.015$  kg MJ<sup>-1</sup> (Wooster et al., 2005), for every 1 MJ of FRE emitted, about  $0.368 \pm 0.015$  kg MJ<sup>-1</sup> (Wooster et al., 2005), for every 1 MJ of FRE emitted, about  $0.368 \pm 0.015$  kg MJ<sup>-1</sup> (Wooster et al., 2005), for every 1 MJ of FRE emitted, about  $0.368 \pm 0.015$  kg MJ<sup>-1</sup> (Wooster et al., 2005), for every 1 MJ of FRE emitted, about  $0.368 \pm 0.015$  kg MJ<sup>-1</sup> (Wooster et al., 2005), for every 1 MJ of FRE emitted, about  $0.368 \pm 0.015$  kg MJ<sup>-1</sup> (Wooster et al., 2005), for every 1 MJ of FRE emitted, about  $0.368 \pm 0.015$  kg MJ<sup>-1</sup> (Wooster et al., 2005), for every 1 MJ of FRE emitted, about  $0.368 \pm 0.015$  kg MJ<sup>-1</sup> (Wooster et al., 2005), for every 1 MJ of FRE emitted, about  $0.368 \pm 0.015$  kg MJ<sup>-1</sup> (Wooster et al., 2005), for every 1 MJ of FRE emitted, about  $0.368 \pm 0.015$  kg MJ<sup>-1</sup> (Wooster et al., 2005), for every 1 MJ of FRE emitted, about  $0.368 \pm 0.015$  kg MJ<sup>-1</sup> (Wooster et al., 2005), for every 1 MJ of FRE emitted, about  $0.368 \pm 0.015$  kg MJ<sup>-1</sup> (Wooster et al., 2005), for every 1 MJ of FRE emitted, about  $0.368 \pm 0.015$  kg MJ<sup>-1</sup> (Wooster et al., 2005), for every 1 MJ of FRE emitted, about  $0.368 \pm 0.015$  kg MJ<sup>-1</sup> (Wooster et al., 2005), for every 1 MJ of FRE emitted, about  $0.368 \pm 0.015$  kg MJ<sup>-1</sup> (Wooster et al., 2005), for every 1 MJ of FRE emitted, about  $0.368 \pm 0.015$  kg MJ<sup>-1</sup> (Wooster et al., 2005), for every 1 MJ of FRE emitted, about  $0.368 \pm 0.015$  kg MJ<sup>-1</sup> (Wooster et al., 2005), for every 1 MJ of FRE emitted, about  $0.368 \pm 0.015$  kg MJ<sup>-1</sup> (Wooster et al., 2005), for every 1 MJ of FRE emitted, about  $0.368 \pm 0.015$  kg MJ<sup>-1</sup> (Wooster et al., 2005), for every 1 MJ of FRE emitted, about  $0.368 \pm 0.015$  kg MJ<sup>-1</sup> (Wooster et al., 2005), for every 1 MJ of FRE emitted, about  $0.368 \pm 0.015$  kg MJ<sup>-1</sup> (Wooster et al., 2005), for every 1 MJ of FRE emitted, about  $0.368 \pm 0.015$  kg MJ<sup>-1</sup> (Wooster et al., 2005), for every 1 MJ of FRE emitted, about  $0.015 \pm 0.015$  kg MJ<sup>-1</sup> (Wooster et al., 2005), for every 1 MJ of FRE emitted, about  $0.015 \pm 0.015$  kg MJ<sup>-1</sup> (Wooster
- 0.015 kg of manzanita DM is consumed. On the other hand, Freeborn et al., (2008) proposed a more
- widely applicable BB coefficient of about 0.453 ± 0.068 kg MJ<sup>-1</sup> through an experimental study of
- different plant fuel types. Therefore, this paper chose to adopt  $0.453 \pm 0.068$  kg MJ<sup>-1</sup> as the biomass
- combustion coefficient to estimate the amount of DM consumed for one hour of combustion at the fire
- point.
- The various emissions generated by biomass combustion can be estimated using the DM consumption
- and the corresponding emission factor (EF). The emission calculation formula is as shown in Eq. (11).

$$E_x = DM_{h,(i,j)} \times EF_x \tag{11}$$

- Where  $E_x(kg)$  represents the emission of substance x (such as CO<sub>2</sub>, CO, NO<sub>x</sub>, etc.) from the fire
- point(i,j) in one hour,  $EF_x$  (g kg<sup>-1</sup>) is the corresponding EF for each substance, which characterizes
- the amount of a specific chemical produced per kilogram of DM burned. Andreae, (2019) conducted a
- comprehensive analysis and compilation of numerous research results. This study selects representative
- emission categories for BB emission estimation. The selected BB EFs for different regions are shown in
- Table 2.

## 2.3.5 Random Forest prediction of SPH and SHAP models

- RF is an ensemble learning method based on the Bagging (Bootstrap Aggregating) principle, proposed
- by Breiman, (2001). Owing to its strong nonlinear modeling capability and scalability, it has been widely
- applied in meteorological and environmental studies to relate atmospheric variables to land surface processes (Üstek et al., 2024; Wang and Wang, 2020). For example, Agrawal et al., (2023) used machine
- learning techniques, along with ERA5 meteorological variables, to build a multivariate regression model

- for wildfire characteristics (such as burned area), successfully predicting the occurrence of large wildfires.
- Moreover, Briggs, (1969) proposed a method for calculating the rise of wildfire plumes based solely on
- buoyancy terms, modeling the heat released by the fire, wind speed, and friction velocity (Haugen, 1982).
- This method is suitable for small-scale wildfires, such as those observed in prescribed burns (Achtemeier
- et al., 2011). However, these methods are limited in their applicability to large-scale wildfires or plume
- rise under complex meteorological conditions (Ferrero et al., 2019). To characterize the relationship
- between wildfire plume rise and meteorological controls, an RF multivariate regression was trained using
- MISR plume heights, ERA5 meteorology, and satellite-derived FRP. The model predicts SPH, which is
- then used to allocate emissions vertically. Following guidance from the Texas Commission on
- Environmental Quality, (2022) and the IS4Fire vertical allocation scheme (Sofiev et al., 2009), 90 % of
- the hourly column emissions are assigned to the upper two-thirds of the predicted plume and 10 % to the
- lower one-third, yielding five vertical layers in total.
- SHAP (Shapley Additive Explanations) is an explanation tool based on game theory, used to quantify the
- contribution of each feature to the predictions of a machine learning model (Mangalathu et al., 2020). By
- calculating the marginal contribution of each feature to the model prediction, SHAP provides
- transparency and interpretability for complex models, such as Random Forest, revealing interactions
- between features (Ekanayake et al., 2022). In this study, in addition to applying the RF model for
- multivariate regression, SHAP was also employed to further analyze the contribution of each
- meteorological variable to SPH.

# 3. Result

333334

349

# 3.1 VIIRS and AHI Data Correction

Figure 4 illustrates the spatial distribution of gridded FRP data derived from VIIRS and AHI, demonstrating the impact of cloud correction algorithms implemented through Eq. (2)-(7). The uncorrected datasets reveal that elevated FRP values are predominantly concentrated within Region 2 for both sensor systems. Due to its superior temporal resolution (10 min revisit time), AHI consistently records higher FRP magnitudes compared to VIIRS. Chen et al., (2022) demonstrated that Himawari-8, with its 2km spatial resolution, detects significantly more fire events than MODIS and VIIRS, consequently yielding elevated FRP measurements. Furthermore, comparisons of thermal anomaly observations from drones with both VIIRS and Himawari-8 data indicate that VIIRS measurements are more reliable. Therefore, this study employs cloud-corrected VIIRS data as a benchmark for calibrating AHI FRP. Region 1, characterized by tropical rainforest (Af) and tropical monsoon (Am) climates, experiences high temperatures, humidity, and frequent rainfall, resulting in extensive cloud cover and frequently underestimated satellite-derived FRP measurements. Prior to cloud correction, the mean VIIRS FRP values in this region are  $11.44 \pm 21.07$  MW and  $11.24 \pm 20.65$  MW in Figure 4 (a) and (d). Following cloud correction, the mean FRP exhibits an approximate 7% increase. Region 2, defined by Am and tropical wet and dry (Aw) climates, is characterized by intense monsoon activity and frequent fire occurrences. After correction, VIIRS FRP increases by 0.6 MW, demonstrating that cloud correction not only mitigates cloud-induced errors but also enhances fire intensity estimation, enabling more accurate detection of fire activity. Regions 3 and 4 similarly exhibit increased VIIRS FRP values following cloud correction, whereas calibrated AHI FRP generally shows a decreasing trend across the study area. However, in regions with substantial BB emissions, such as northern Laos, AHI FRP still

- increases, likely as cloud correction reveals additional fire activity, thereby yielding higher observed FRP
- values.

357358

360361

385

387

389

#### 3.2 Reconstruction of the FRP daily cycle

## 3.2.1 Gaussian-based fitting of FRP diurnal climatology

To establish a robust foundation for accurately filling temporal gaps in BB emission inventories, climatological diurnal FRP cycles were fitted using Gaussian functions. Historical Himawari-8/9 (AHI) FRP observations from 2016 to 2023 were used to derive climatological diurnal cycles for four representative regions (Regions 1-4) and five vegetation types (cropland, forest, grassland, peatland, and shrubland) in Figure S4. The Gaussian fitting performed well in most regions and vegetation types with a mean R<sup>2</sup> exceeding 0.87, confirming its effectiveness in capturing diurnal FRP variability. However, significant differences in diurnal patterns were observed across regions and vegetation types. Region 1 exhibited relatively lower fire intensity but still displayed clear unimodal diurnal patterns, peaking from morning to midday (local time), possibly related to agricultural practices or sustained peatland fires. Region 2 displayed pronounced afternoon peaks, particularly evident in grassland ( $R^2 = 0.98$ ). In Regions 3 and 4, cropland, forest, and grassland showed excellent fitting performance ( $R^2 \ge 0.91$ ), reflecting distinct anthropogenic burning patterns. For instance, cropland fires in Northeast China exhibited clear unimodal diurnal cycles, predominantly concentrated between 9:00 and 16:00 local time. These Gaussian function-based climatological FRP diurnal cycles effectively characterize the typical diurnal fire variations across different regions and vegetation types, establishing essential groundwork for further developing dynamic gap-filling methods and improving the continuity and reliability of satellite-derived fire observations.

## 3.2.2 Dynamic adjustment and gap-filling of FRP diurnal cycles

To enhance the spatiotemporal accuracy and reliability of BB emission inventories in SEA, a regionally adaptive approach was developed for dynamic adjustment and climatological gap filling of FRP based on region-specific observational characteristics. Using long-term AHI observations, climatological diurnal FRP cycles were reconstructed and applied to representative fire events in 2023 across four key regions and five vegetation types (Figure 5). The Gaussian Least Squares (GLS) fitting consistently delivered robust performance across all regions and ecosystems, with coefficients of determination (R2) reaching up to 0.98, confirming the reliability and broad applicability of the method for daily-scale FRP reconstruction. In Region 1, cropland fires exhibited a distinct and well-captured morning peak (UTC 02:00-06:00, approximately 09:00-13:00 local time) that was well captured by the dynamic fitting, achieving high accuracy ( $R^2 = 0.82$ ) and strong agreement between the fitted curves and observations. In Region 2, dynamic Gaussian fitting methods, including GLS and Gaussian Vertical Movement (GVM), substantially outperformed conventional climatological fitting, underscoring their advantage in reproducing actual fire behavior. Regions 3 and 4 also demonstrated strong fitting results for forest and grassland fires, reflecting highly regular diurnal fire patterns and the effectiveness of dynamic parameter adjustment in identifying peak burning periods. Notably, in Northeast China, all vegetation types except grasslands, which lacked sufficient observations for fitting, exhibited the highest fitting performance (R<sup>2</sup> ≥ 0.8), with cropland fires showing a clearly defined unimodal diurnal pattern. This reflects the influence of well-regulated anthropogenic burning activities, such as crop residue combustion, or seasonally managed fire regimes during official fire prevention periods, underscoring the strong temporal

406

411

423

434

436

regularity of fires in this region and further validating the reliability of the proposed approach.

The dynamic gap-filling algorithm substantially improved both the temporal continuity and quantitative accuracy of FRP diurnal cycles, effectively mitigating observational deficiencies caused by cloud contamination and the spatiotemporal sampling limitations of satellite-based fire detection (Figure 6). Across all regions and vegetation types, reconstructed FRP profiles showed marked enhancements, particularly during key burning periods underrepresented in the original observations. In Region 1, cropland fires exhibited pronounced morning peaks (08:00-12:00 local time), with FRP increased by 67.7% after reconstruction, consistent with the common practice of morning crop residue burning. Shrubland fires in the same region showed a maximum enhancement of 80.6%, indicating active morning burning in tropical shrublands that was systematically underdetected in the original data. In Region 2, cropland FRP peaks during the afternoon and early evening (14:00-20:00 local time) increased by 74.2%, reflecting traditional afternoon burning practices, while overall FRP corrections were greater than in other regions. Forest fires in this region showed a 53.9% enhancement between 12:00 and 14:00, and grassland fires increased by 68.5% between 12:00 and 18:00, both highlighting intensified daytime combustion under dry conditions. In Region 3, forest and shrubland fires exhibited the highest enhancements, reaching up to 82.4%, whereas peatland fires had the lowest adjustments (25.4%), consistent with their stable smoldering characteristics and weak diurnal variability. In Region 4, shrubland FRP increased by 88.7% during 16:00-00:00 local time, revealing active evening-to-night burning that was systematically underestimated due to twilight detection gaps, with cropland fires showing the largest absolute increase (approximately 1 × 10<sup>6</sup> W m<sup>-1</sup>). Overall, the dynamically adjusted FRP profiles exhibited markedly improved temporal continuity compared with the original observations, capturing the primary peaks of fire activity and recovering missing signals during under-sampled periods, particularly for long-duration events ( $\geq$ 3 h), thereby providing a more accurate temporal representation essential for high-resolution emission modeling and atmospheric transport simulations.

Figure 7 compares the mean daily FRP distributions across SEA in 2023 under three scenarios: (a) the dynamic adjustment and gap-filling method, (b) the original observations without Gaussian fitting, and (c) conventional Gaussian fitting. The original observations systematically underestimate FRP due to cloud contamination, low temporal sampling frequency, and twilight detection blind zones. This underestimation is evident across all key regions, with low regional mean FRP values (21.34 MW in Region 1, 23.04 MW in Region 2, and 17.21 MW in Regions 3-4) and large standard deviations, indicating high spatiotemporal variability and missing peak fire activity (Figure 7 (b)). While the Gaussian fitting method improves spatial completeness by reconstructing climatological FRP diurnal curves, it neglects actual temporal fire dynamics, resulting in systematic overestimation in certain regions. For example, Region 2 showed a 2.17% increase in FRP relative to the original observations (23.35 MW), primarily due to artificial amplification during inactive periods (Figure 7 (c)). In contrast, the dynamically adjusted method incorporates region- and vegetation-specific diurnal characteristics, such as cropland burning peaks in the afternoon and shrubland fires occurring during twilight hours, leading to more realistic and continuous reconstructions. The dynamically fitted FRP showed improved regional means (e.g., 23.85 MW in Region 2 and 17.49 MW in Region 4), representing relative increases of 3.52% and 1.62% compared to the original data. In addition, the dynamic method effectively recovers underdetected fire signals, particularly in Regions 2 and 4 (0.81 MW and 0.23 MW, respectively), while also avoiding the overestimation seen in conventional Gaussian fitting methods (e.g., -0.51 MW in

- Region 2). These results demonstrated that the proposed approach enhances both the accuracy and
- representativeness of FRP spatial distributions by capturing realistic fire peaks and avoiding artificial
- inflation during inactive hours, thus providing a more reliable input for high-resolution emission
- modeling.

450

## 3.3 Comparison 2D BB with TROPOMI CO and five inventories

- In this study, a top-down BB emission inventory for SEA was developed using a sequential conversion
- framework (Eqs. 9-11) from FRP to fire radiative energy (FRE), dry matter (DM) consumption, and
- ultimately to pollutant emissions. The FRP input was obtained from dynamically reconstructed diurnal
- cycles (with enhanced spatiotemporal continuity). FRE was calculated by integrating the hourly FRP
- series, providing a quantitative measure of total fire energy release. DM consumption was estimated from
- FRE using a biomass combustion coefficient of 0.453 ± 0.068 kg MJ<sup>-1</sup>. Pollutant emissions, including
- CO<sub>2</sub>, CO, nitrogen oxides (NO<sub>x</sub>), PM<sub>2.5</sub>, organic carbon (OC), and BC, among others, were subsequently
- calculated by applying vegetation-specific emission factors (EFs, Table 2).

#### 3.3.1 Satellite-based evaluation of SEAF CO emissions

- Figure S5 displays the monthly mean distribution of CO column concentrations retrieved from
- TROPOMI over SEA for 2023, revealing a pronounced seasonal enhancement during the spring burning
- season (March-April). Notably elevated values, exceeding 0.08 mol m<sup>-2</sup>, are observed over northern
- Myanmar, northern Thailand, and western Laos. In comparison, Figure 8 (b-m) shows the monthly mean
- CO emissions derived from the SEAF inventory, which exhibit remarkably consistent spatial and
- temporal patterns with the satellite observations. The Region 2 shows a distinct emission peak during
- March and April, with maximum hourly emissions exceeding  $0.8 \times 10^6$  g h<sup>-1</sup>, closely matching the spatial
- extent and intensity of TROPOMI-observed CO enhancements. Moreover, SEAF emissions also
- captured the temporal evolution of CO concentrations with high fidelity. The monthly SEAF-derived CO
- emissions in Region 2 exhibit a strong linear correlation with TROPOMI CO column densities (R = 0.97)
- in Figure 8 (a). Both datasets reflect a coherent seasonal trend: a progressive increase from January to
- March, a clear peak in March, followed by a substantial decline through September, and a modest
- rebound toward the end of the year. Importantly, the SEAF inventory not only reproduced the seasonal
- variability but also successfully captured the precise timing and magnitude of the peak fire season.
- To assess the accuracy of the SEAF inventory at the event scale, a representative BB episode that
- occurred on 9 March 2023 was examined using multi-source satellite data (Figure 9). The fire location
- and associated smoke plume evolution were clearly captured by VIIRS (Figure 9 (a)) and time-resolved
- Himawari-9 true-color imagery (Figure 9 (c-n)), with red markers indicating active fire pixels.
- Corresponding CO emissions were quantified from both TROPOMI satellite retrievals and the SEAF
- inventory (Figure 9 (b)). The SEAF-derived CO emissions for this event totaled 0.307 Gg, closely
- aligning with the TROPOMI-based estimate of 0.283 Gg. The relative deviation of 7.81% was well
- within the  $\leq$ 10% random error margin defined for the TROPOMI CO product (Martínez-Alonso et al.,
- 2020), demonstrating the inventory's strong capacity to reproduce fire-induced emissions from individual
- events with high accuracy.

# 3.3.2 Comparison of SEAF-derived PM<sub>2.5</sub> with five existing BB inventories

To evaluate the reliability of PM<sub>2.5</sub> emissions estimated by the SEAF inventory, a quantitative comparison

480

482

484

was performed against five widely used BB emission inventories (GFAS, FINN, FEER, QFED, and IS4FIRES). The SEAF inventory showed a total annual PM<sub>2.5</sub> emission of 2362 Gg yr<sup>-1</sup> over SEA in 2023, which lies near the midpoint among the values given by the selected inventories in Figure 10 (a). This value is substantially lower than that of FINN v2.5.1 (7099 Gg yr<sup>-1</sup>), which has been shown to overestimate BB emissions in this region (Jin et al., 2024). Relative to FINN, SEAF reduced the estimated emissions by approximately 66.7%. Emission estimates from SEAF closely aligned with those of FEER v1.0 (2335 Gg yr<sup>-1</sup>) and QFED v2.6r1 (2345 Gg yr<sup>-1</sup>), suggested that the SEAF estimates are constrained and consistent with other satellite-derived products. Moreover, SEAF showed strong consistency with FEER and QFED in Regions 1 and 2. In contrast, FINN consistently produces higher estimates, with the largest discrepancy in Region 2, where its emissions are nearly four times those of SEAF (Figure 10 (b-e)). Notably, SEAF also excelled in capturing the seasonal variability of PM<sub>2.5</sub> emissions (Figure S6). In Region 2, emissions peak during March and April, reaching approximately 500 Gg month<sup>-1</sup>, consistent with dry-season fire activity. This peak was reproduced by SEAF through a dynamic diurnal gap-filling approach that reconstructs temporal fire intensity patterns. GFAS and FEER underestimated the seasonal maximum, while FINN overestimated emissions and did not accurately reflect seasonal trends.

Regarding spatial distribution, SEAF accurately delineated key emission hotspots over northern Myanmar, northern Thailand, and western Laos (Figure 11), showed strong agreement with observed CO column enhancements from TROPOMI. In contrast, GFAS and FEER generated more spatially diffuse and inconsistent patterns, while FINN tended to overestimate both the magnitude and spatial extent of emissions across SEA. Spatial resolution also contributed significantly to inventory performance (Figure S7). SEAF (3 km) and FINN (1 km) provided finer-scale spatial detail compared to the coarser 10 km resolution of GFAS, QFED, FEER, and IS4FIRES. SEAF demonstrated enhanced spatial fidelity, effectively capturing localized emission hotspots and surface heterogeneity, including water bodies and bare land, particularly in topographically complex regions. While FINN showed similarly fine spatial resolution, it frequently overestimated emissions across various regions, resulting in higher total emissions and exaggerated spatial coverage. In contrast, coarse-resolution inventories smooth localized features, potentially obscuring critical emission signals.

# 3.4 Prediction of smoke plume height

506 Figure 12 (a) presents the SPH predicted by the RF model, demonstrating a high overall consistency with the MISR observations ( $R^2 = 0.9$ , RMSE = 334.68 m). Predictions falling within the reasonable range 507 (defined as "Good", with a bias within ±500 m) accounted for 90.6% of the RF model results. In contrast, 508 509 the traditional IS4FIRES achieved predictions within the "Good" range for only 57% of cases, while the PRM scheme commonly employed in air quality models performed even lower, at merely 44% (Rémy 510 511 et al., 2017). Furthermore, the RMSE values for the traditional models were significantly higher, at 533 m and 955 m, respectively, compared to the 334.68 m RMSE of the RF model developed in this study. 512 513 These results collectively indicate a substantial advantage of the present machine learning approach for 514 predicting SPH.

SHAP analysis was applied to interpret the contribution of environmental variables to SPH predictions, 516 providing insights into both the magnitude and direction of each factor's influence. The SHAP value sign

indicates whether a variable positively or negatively affects SPH, while color represents the variable's

521

531532

magnitude (red for high, blue for low). Temperature- and radiation-related variables emerged as the dominant drivers (Figure 12 (b)), consistent with previous findings that atmospheric temperature governs the buoyant transport of BB plumes (Feng et al., 2024; Freitas et al., 2007; Val Martin et al., 2010). In particular, the vertical integral of temperature (Vit) was the most significant factor, capturing the effect of atmospheric thermal structure on plume rise: larger vertical temperature gradients provide greater buoyant energy, leading to higher injection altitudes. Terrain elevation (z) was also identified as a key factor, as elevated regions promote stronger localized convection, especially in topographically complex areas like SEA. Longitude demonstrated high importance as well, reflecting the east-west climatic and geographic heterogeneity that significantly influences plume dynamics. Other notable variables included surface solar radiation (ssr), month, and latitude. Previous studies (Cohen et al., 2018; Feng et al., 2024; Holanda et al., 2023) have shown that seasonal variations in surface heating and fire activity during dry periods can enhance plume rise. The RF-SHAP model further revealed that these seasonal variables interact in a complex and nonlinear manner, particularly involving month, solar radiation, surface heating, and fire intensity. This intricate interplay helps explain why traditional models tend to perform poorly in regions characterized by strong seasonal variability. Additional factors such as planetary boundary layer height (blh), 10 m wind speed (v10), and FRP also contributed substantially. Higher FRP increases the mechanical energy available for vertical transport, while elevated PBL height offers a channel for plume penetration into the free troposphere. Importantly, SHAP analysis revealed that the influence of FRP is highly dependent on meteorological conditions, exhibiting strong nonlinear relationships that are often oversimplified in traditional models such as PRM. Although variables such as near-surface humidity (d2m), vegetation index (lai hv), and sensible heat flux (sshf) played relatively minor roles compared to dominant predictors, they still contributed meaningful information related to aerosol microphysics, fuel availability, and surface energy exchange. The RF-SHAP framework effectively integrates these nonlinear and region-dependent factors, providing a more comprehensive and interpretable alternative to conventional plume-rise schemes.

#### 3.5 3D Biomass Burning inventory assessment

- Based on the previously constructed 2D SEAF inventory and SPH derived from the RF-SHAP model, a
- high-resolution 3D BB emission inventory (3D SEAF) was constructed in this study. Vertical allocation
- followed the approach proposed by the Texas Commission on Environmental Quality and the five-layer
- scheme of IS4FIRES (0.025km, 0.275km, 1.0km, 2.75km, and 5.5km) (Texas Commission on
- Environmental Quality, 2022). This approach yields a vertically resolved PM<sub>2.5</sub> emission dataset across
- five altitude bands.
- A comparison of monthly emissions from the 3D SEAF and IS4FIRES inventories (Figure 13) reveals
- similar seasonal patterns, with both inventories capturing a pronounced peak in fire emissions during
- March and April. Both inventories show a pronounced vertical uplift during these months, reflecting the
- intense burning and consequent plume rise in the dry season. Notably, during the peak period (March-
- April), SEAF allocates less PM<sub>2.5</sub> to the lowest layers (0.025 km and 0.275 km) than IS4FIRES, while
- substantially increasing emissions in the upper layers (2.75 km and 5.5 km). This suggests that IS4FIRES
- may underestimate upper-level emissions, whereas SEAF offers a distribution more consistent with
- MISR-observed plume structures.
- Figure 14 (a-e) illustrates the spatial distribution of 3D SEAF emissions across different vertical layers.

561562

564565

566567

At the lowest level (0.025 km), emission intensities are generally low, with pronounced hotspots primarily located in northern Myanmar, northern Thailand, and Laos, where values reach up to 0.4 g m <sup>2</sup> yr<sup>-1</sup>. With increasing altitude, particularly at 0.275 km and 1 km, emission intensities increase significantly, with peak values of approximately 1.6 g m<sup>-2</sup> yr<sup>-1</sup> observed across Region 2. As altitude increases further to 2.75 km and 5.5 km, emission hotspots become increasingly confined, and the spatial extent of high-emission areas is reduced. Although the emission intensity at 5.5 km decreases relative to lower layers, notable localized plumes persist, especially over northern Laos. These patterns underscore the characteristic vertical uplift of BB plumes, extending from the near-surface to the lower troposphere. Figure 14 (f) further compares the vertical frequency distribution of SEAF emissions with those from MISR, GFAS, IS4FIRES, and CALIPSO satellite observations across SEA. The SEAF inventory exhibits a strong peak in emission frequency below 1km, reaching a maximum relative frequency of ~0.7, followed by a rapid decline above 1 km. Nevertheless, SEAF still registers non-negligible emissions above 2.75 km, reflecting its ability to represent both surface-concentrated and elevated plume injection events. This vertical profile closely aligns with CALIPSO observations, which also reveal near-surface dominance in aerosol vertical structure. In contrast, MISR, GFAS, and IS4FIRES display a broader vertical distribution of emissions. Specifically, GFAS exhibits relatively high emission frequencies in the 3-5 km altitude range, while MISR and IS4FIRES maintain substantial emission fractions between 2.75 and 5.5 km. Although the SEAF inventory shows lower emission frequencies in the middle and upper atmospheric layers compared to these inventories, it still retains a persistent, albeit smaller, fraction of emissions at 5.5 km. Notably, this aligns well with the extended plume tails observed by CALIPSO and GFAS, indicating the SEAF inventory's ability to represent both the near-surface concentration of BB plumes and the occurrence of elevated smoke layers. Such performance is consistent with independent satellite observations and highlights the realistic representation of plume dynamics provided by the SEAF vertical allocation scheme.

# 4. Discussion

The SEAF BB emission inventory developed in this study advances spatiotemporal resolution, dynamic adjustment, and vertical distribution modeling. Cloud correction and cross-calibration between VIIRS and Himawari-8/9 reduce biases associated with cloud cover and revisit cycles, yet uncertainties remain under extreme meteorological conditions and at large satellite zenith angles where simple cloud-fraction metrics cannot fully capture fire variability (Wang et al., 2018; Xie et al., 2018). The reconstruction of FRP diurnal cycles through Gaussian fitting effectively addresses data gaps but assumes a unimodal daily pattern. This simplification does not always reflect BB activity in SEA, where agricultural burning, peatland fires, and anomalous climate events often produce bimodal or irregular temporal structures (Fan et al., 2023; Yin, 2020). Regarding peatland fires, we acknowledge the inherent limitation of the FRP-derived top-down approach in capturing emissions from deep smoldering combustion. While this study seeks to address this limitation by applying peatland-specific emission factors, a strategy that yields regional totals in broad agreement with other inventories, the potential for underestimation remains a key source of uncertainty (Fisher et al., 2020). Consequently, rapid fluctuations or emergent fire behaviors may be underestimated despite the application of dynamic adjustments.

For the vertical allocation, a RF-SHAP model trained with MISR plume heights, ERA5 meteorology, and FRP was used to predict SPH, which then guided a five-layer distribution scheme following the Texas Commission on Environmental Quality and IS4FIRES. This hybrid approach links data-driven

SPH prediction with a structured allocation framework and offers advantages over conventional plume-602 rise parameterizations. However, the coarse resolution of ERA5 together with the sparse sampling of MISR limit the representation of fine-scale convection and extreme lofting, resulting in potential 603 604 underestimation during localized outbreaks (Sessions et al., 2011; Val Martin et al., 2012). Compared with GFAS, FINN, FEER, QFED, and IS4FIRES, SEAF captures emission magnitudes and seasonal 605 variability more accurately, yet short-lived peaks and near-surface emissions remain underrepresented, 606 607 reflecting the emphasis on dominant injection layers rather than rare extreme events. Further 608 improvements will require higher-resolution meteorological fields, integration of additional 609 geostationary platforms such as Geostationary Environment Monitoring Spectrometer (GEMS) and 610 GOES-R, and complementary lidar observations (e.g., CALIPSO) to better constrain vertical injection 611 under extreme and under-sampled conditions.

#### 5. Conclusion

- The SEAF emission inventory was developed as an hourly 3 km resolution 2D/3D dataset for 2023,
- addressing deficiencies in diurnal profiles and vertical injection of BB emissions. The inventory
- integrates cloud-corrected, cross-calibrated FRP from AHI and VIIRS with a region- and vegetation-
- specific Gaussian reconstruction and dynamic gap filling, restoring missing peaks while minimizing
- artificial inflation. Validation against TROPOMI CO (R = 0.97) and independent estimates indicates high
- reliability, with annual PM<sub>2.5</sub> emissions (2362 Gg) consistent with FEER and QFED and substantially
- lower than FINN. The vertical dimension is constrained through a RF-SHAP interpretation trained with
- MISR and ERA5, achieving an  $R^2 = 0.90$  and an RMSE = 335 m, and reallocating injection from near
- surface layers toward 2.75 and 5.5 km during the spring burning peak in closer accordance with MISR and CALIPSO structures. These improvements in temporal completeness and vertical realism enhance
- the representation of BB emissions in chemical transport models, supporting more robust assessments of
- air quality, transboundary smoke transport, and radiative impacts in SEA.

#### 625 Data Availability

- The SEAF emission inventory developed in this study, including 2D/3D hourly products at 3 km
- resolution for 2023, is publicly available at Zenodo (https://doi.org/10.5281/zenodo.16793129) (Jin et
- al., 2025). Satellite datasets used include FRP from the AHI onboard Himawari-8/9 provided by the Japan
- Meteorological Agency (JMA), FRP from the VIIRS onboard Suomi-NPP and NOAA-20 provided by
- NASA/NOAA, column carbon monoxide (CO) from the TROPOMI operated by the European Space
- Agency (ESA), and plume height observations from the MISR provided by NASA. ERA5 meteorological
- reanalysis data were obtained from the ECMWF. All datasets are openly accessible from their respective
- providers.

### 634 Competing interests

The authors declare that they have no conflict of interest.

# 636 Acknowledgments

- The authors gratefully acknowledge the Japan Meteorological Agency (JMA) for Himawari-8/9 AHI data,
- NASA/NOAA for VIIRS FRP products, the European Space Agency (ESA) for TROPOMI CO data,
- NASA for MISR plume height observations, and the European Centre for Medium-Range Weather

# https://doi.org/10.5194/essd-2025-515 Preprint. Discussion started: 21 October 2025 © Author(s) 2025. CC BY 4.0 License.

| 640 | Forecasts (ECMWF) for ERA5 meteorological data. This work was supported by the Science and     |
|-----|------------------------------------------------------------------------------------------------|
| 641 | Technology Program of Guangdong Province (Science and Technology Innovation Platform Category) |
| 642 | (Grant No. 2019B121201002), and the National Natural Science Foundation of China (Grant No.    |
| 643 | 42075181, 42375182, and 42175086). Innovation Group Project of Southern Marine Science and     |
| 644 | Engineering Guangdong Laboratory (Zhuhai) (No. 311024001).                                     |
|     |                                                                                                |

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

# https://doi.org/10.5194/essd-2025-515 Preprint. Discussion started: 21 October 2025 © Author(s) 2025. CC BY 4.0 License.

(a)

| 910 | J. W., Wiedinmyer, C., and Da Silva, A.: Sensitivity of mesoscale modeling of smoke direct radiative        |
|-----|-------------------------------------------------------------------------------------------------------------|
| 911 | effect to the emission inventory: a case study in northern sub-Saharan African region, Environ. Res. Lett., |
| 912 | 9, 075002, https://doi.org/10.1088/1748-9326/9/7/075002, 2014.                                              |
| 913 | Zhang, X., Kondragunta, S., Ram, J., Schmidt, C., and Huang, H.: Near-real-time global biomass burning      |
| 914 | emissions product from geostationary satellite constellation, J. Geophys. Res.: Atmos., 117,                |
| 915 | 2012JD017459, https://doi.org/10.1029/2012JD017459, 2012.                                                   |
| 916 | Zhu, L., Val Martin, M., Gatti, L. V., Kahn, R., Hecobian, A., and Fischer, E. V.: Development and          |
| 917 | implementation of a new biomass burning emissions injection height scheme (BBEIH v1.0) for the              |
| 918 | GEOS-chem model (v9-01-01), Geosci. Model Dev., 11, 4103-4116, https://doi.org/10.5194/gmd-11-              |
| 919 | 4103-2018, 2018.                                                                                            |
| 920 |                                                                                                             |
| 921 |                                                                                                             |
| 922 |                                                                                                             |
| 923 |                                                                                                             |
| 924 |                                                                                                             |
| 925 |                                                                                                             |
| 926 |                                                                                                             |

Table 1. Statistics on the duration of sustained burning and periods of high probability of burning
 in different regions and vegetation types (Figure 1).

| Regions | Vegetation | T_gap (hours) <sup>a</sup> | Filling periods (UTC) |
|---------|------------|----------------------------|-----------------------|
| 1       | Croplands  | 3                          | 1-3                   |
| 1       | Forests    | 4                          | 2-5                   |
| 1       | Grasslands | 3                          | 1-3                   |
| 1       | Peatlands  | 3                          | 4-6, 20-22            |
| 1       | Shrublands | 10                         | 1-10                  |
| 2       | Croplands  | 7                          | 7-13                  |
| 2       | Forests    | 3                          | 5-7                   |
| 2       | Grasslands | 7                          | 5-11                  |
| 2       | Peatlands  | 3                          | 21-23                 |
| 2       | Shrublands | 3                          | 5-7                   |
| 3       | Croplands  | 3                          | 6-8                   |
| 3       | Forests    | 10                         | 0-9                   |
| 3       | Grasslands | 3                          | 1-3                   |
| 3       | Peatlands  | 3                          | 20-22                 |
| 3       | Shrublands | 9                          | 0-8                   |
| 4       | Croplands  | 5                          | 7-11                  |
| 4       | Forests    | 6                          | 5-10                  |
| 4       | Grasslands | 14                         | 0-13                  |
| 4       | Peatlands  | 10                         | 13-22                 |
| 4       | Shrublands | 3                          | 3-5                   |

aT\_gap: longest continuous fire duration within a high-burning period (frequency  $\geq 0.9$ ), allowing merging if separated by short gaps (mean frequency  $\geq 0.5$ ).

# 931 Table 2. Emission factors (unit: g kg<sup>-1</sup>)

| Species           | Forest | Shrubland, Grassland | Cropland | Peatland |
|-------------------|--------|----------------------|----------|----------|
| CO <sub>2</sub>   | 1570   | 1660                 | 1430     | 1590     |
| СО                | 113    | 69                   | 76       | 260      |
| $NO_x$            | 3.0    | 2.5                  | 2.4      | 1.2      |
| NH <sub>3</sub>   | 0.98   | 0.89                 | 0.99     | 4.2      |
| $SO_2$            | 0.70   | 0.47                 | 0.80     | 4.3      |
| PM <sub>2.5</sub> | 18.5   | 6.7                  | 8.2      | 18.9     |
| OC                | 10.9   | 3.0                  | 4.9      | 14.2     |
| BC                | 0.55   | 0.53                 | 0.42     | 0.10     |

Figure 1. (a) MODIS land cover for 2023 in Southeast and East Asia. (b) Köppen climate classification Map (climate baseline 1991-2020), with representative high biomass burning emission regions based on Giglio et al., (2006), including 1. Southern Southeast Asia, 2. Mainland Southeast Asia, 3. Southern China, and 4. Northern China. Climate types include Af (tropical rainforest), Am (tropical monsoon), Aw (tropical savanna), Bwk (cold desert), Bsk (cold semi-arid), Cwa (humid subtropical with dry winter and hot summer), Cwb (temperate highland tropical climate with dry winter and warm summer), Cfa (humid subtropical with hot summer and no dry season), Dwa (humid continental with dry winter and hot summer), Dwb (humid continental with dry winter and cold summer), and ET (tundra).

958

959

Figure 2. Methodological framework for constructing hourly three-dimensional biomass burning emission inventories in Southeast and East Asia.

Figure 3. Characterizes the temporal distribution of sustained burning  $\geq 3$  hours in different regions and land types.

Figure 4. Cloud correction of gridded FRP data from VIIRS and AHI. The first row shows the correction for NPP, the second row for NOAA20, and the third row for Himawari-9. The last column displays the difference between the corrected and uncorrected FRP data.

Figure 5 Dynamically adjusted Gaussian gap-filling of FRP diurnal cycles for representative regions and vegetation types in 2023. Solid black circles represent original observed FRP data, solid red triangles represent gap-filled FRP values for missing observations, gray solid lines are original climatological Gaussian fitting, blue dash-dot lines show dynamically adjusted Gaussian fitting results (Gaussian Least Squares, GLS), green dotted lines represent climatological Gaussian fitting with adjustment factor *d* only (Gaussian Vertical Movement, GVM).

Figure 6. Bar chart comparing the total hourly FRP (2023) before and after Gaussian fitting adjustments across four climatic regions and five vegetation types. The time of the maximum proportion of filled FRP to unfilled FRP is also annotated, with some panels displaying enlarged insets.

Figure 7 Spatial comparison of mean daily fire radiative power (FRP) distributions in the SEA region during 2023. (a) gap-filled dynamic reconstruction, (b) original observations, (c) traditional Gaussian fitting, (d) difference between (a) and (b), and (e) difference between (a) and (c).

Figure 8 Monthly CO emissions over SEA in 2023 based on the SEAF inventory and comparison with satellite observations. (a) Temporal variation of monthly CO emissions from SEAF and CO column concentrations from TROPOMI over Region 2; (b-m) Spatial distribution of monthly mean CO emissions derived from SEAF.

Figure 9 (a) True-color image from VIIRS, (b) comparison between CO emissions from TROPOMI observations and the SEAF emission inventory, and (c) true-color image from Himawari-9, with red dots indicating fire locations.

1017

1018

1019

1020

Figure 10 Comparison of SEAF PM<sub>2.5</sub> emissions with five BB emission inventories.

Figure 11 Spatial comparison of  $PM_{2.5}$  emissions from SEAF and five BB emission inventories.

1034 1035

Figure 12 (a) Random Forest (RF)-based prediction of BB plume height and (b) SHAP-based analysis of key driving variables (Table S1).

Figure 13 Vertical distribution comparison of SEAF and IS4FIRES PM<sub>2.5</sub> (Jan-Dec) emissions.

Figure 14 (a-e) Spatial distribution of SEAF  $PM_{2.5}$  emissions at vertical five altitude levels (0.025-5.5 km), and (f) vertical comparison.

1049 1050