# Peer review of "Three-Dimensional Biomass Burning Emission Inventory for"

_Earth System Science Data, 2025_

## Referee Comment (RC2)

This paper generates three-dimensional biomass-burning emissions for Southeast and East Asia by developing new fire diurnal cycles and vertical injection profiles. The proposed diurnal cycle is derived by integrating fire radiative power data from both geostationary and polar-orbiting satellite observations. The vertical injection profile is produced using a machine-learning model trained on satellite-retrieved smoke plume heights and meteorological variables. However, the manuscript is not well-structured, and the methodology lacks clarity and robustness. Substantial revisions are needed before the work can be considered for publication.

**Major comments:**
1. **Weak integration between the fire diurnal cycle and vertical injection profile components**
   Although the overarching objective is to develop a three-dimensional biomass-burning emission dataset, the manuscript presents the derivation of the fire diurnal cycle and the vertical injection profile as largely independent processes. The authors first generate and validate 2-D fire emissions, then separately develop and validate smoke plume heights. However, the connection between these two components—and how they integrate to form the final 3-D emissions—is not clearly articulated. I recommend extensively restructuring the manuscript to better emphasize the methodological coherence and the interdependencies between these two parts.

2. **Insufficient background and regional context**
   The literature review does not adequately cover relevant studies conducted in Asia. Much of the discussion focuses on work from the United States or other regions, while several closely related studies in Asia are overlooked. These omissions weaken the contextual grounding of the study and obscure how this work builds upon or differs from existing regional research. The authors should expand the background section to include key Asian studies and clearly articulate their linkages to the current work.

   References:
   'Dynamics of fire plumes and smoke clouds associated with peat and deforestation fires in Indonesia'
   'Fire Particulate Emissions from Combined VIIRS and AHI Data for Indonesia, 2015-2020'
   'Improved estimation of fire particulate emissions using a combination of VIIRS and AHI data for Indonesia during 2015–2020'
   'Highly anomalous fire emissions from the 2019–2020 Australian bushfires'

3. **Methodological issues and lack of robustness**
   - **Choice of VIIRS 375-m product:**
     The manuscript uses 375-m VIIRS FRP, but this product is known to saturate under high-intensity fires. Why was the 375-m product chosen instead of the 750-m VIIRS FRP? This choice needs stronger justification beyond the higher spatial resolution. Have the authors tested or compared the 750-m FRP?
   - **AHI FRP correction:**
     The reference (Li et al., 2022) cited for the AHI FRP correction does not actually use AHI FRP, making this citation inappropriate.

- **Unclear machine-learning description:**
  The section describing the machine-learning method lacks clarity. Important details such as the training–testing strategy, sample selection, and dataset composition are not provided.
- **Missing validation subsection in the Methods:**
  A dedicated validation subsection should be added to the Methods section to clearly explain the evaluation workflow.
- **Lack of GFED products in evaluation:**
  It is unclear why the widely used GFED-related products are omitted from the evaluation. Including them would provide a more comprehensive comparison.
- **Section 3.3.2 — Rationale for comparison with inventories:**
  The manuscript evaluates the new emissions against existing emission inventories but mainly describes the differences without explaining the underlying causes. For example, why are the results lower than FINN but closer to FEER and QFED? Additional interpretation is needed in the Results or Discussion sections.
- **Lines 321–324 — Transferability of methods:**
  These lines describe a core component of the method, yet the supporting references are based on studies from the U.S. and Europe. The authors should justify whether such methodologies are appropriate for Asian fire regimes.
- **Line 465 — Limited validation case:**
  Validating the method using only a single biomass-burning episode is insufficient. More cases are needed to demonstrate robustness.

**4. Errors in basic information**
There are several factual inaccuracies that need correction. For example:
- Mixing up sensor and satellite names (line 97).
- Stating that MODIS fails to capture nighttime events (lines 131–132).
- Citing a reference for AHI FRP correction that does not actually use AHI FRP.

**Minor to moderate comments:**
- **Lines 33–34:** Please specify the versions of all datasets used—at minimum in the Data section—since different versions may produce substantially different values.
- **Lines 33–34:** Why not provide comparison results for all emission inventories included in the study?
- **Line 35:** Please use statistical metrics to demonstrate the performance of the SPH estimation rather than presenting a single numerical value.
- **Line 38:** "Satellite observations" should be specified. If you mean MISR SPH, please explicitly name the product here.
- **Lines 38–41:** It seems inconsistent that the study's final goal is to generate 3-D fire emissions, yet this section focuses only on analyzing drivers of SPH.
- **Line 41:** The phrase "are anticipated to" is not appropriate, since you have already generated an observation-driven, hourly 3-D biomass-burning emission dataset for SEA.

- **Lines 47–51:** If the manuscript focuses on validating CO, why not center the discussion on CO emissions in this section?
- **Lines 56–58:** The justification for selecting the SEA study area is not strong or straightforward.
- **Lines 56–81:** The background research for Asia is insufficient. Much prior work outside Asia is discussed, yet several important regional studies are missing.
- **Line 95:** Please add "three-dimensional" here for accuracy.
- **Line 97:** VIIRS and NOAA-20 should not be presented as parallel entities; one is a sensor, the other is a satellite platform.
- **Lines 102–103:** Please remove the final sentence—its style is more appropriate for a proposal rather than a manuscript.
- **Lines 129–130:** The description of Himawari-8's spatial and temporal resolution is unnecessary here, as these specifics have already been provided.
- **Lines 131–132:** MODIS can detect nighttime events due to its ~1:30 a.m. overpass, although it may not capture all nighttime fires. This statement should be corrected.
- **Lines 163–165:** Please clarify what constitutes the testing dataset.
- **Lines 171–174:** Why not directly include the common fire weather index variables (temperature, relative humidity, wind, and precipitation)?
- **Lines 177–180:** Why are widely used GFED products not included?
- **Section 2.3.1:** If this section refers to VIIRS FRP data calibration, please revise the title to reflect that.
- **Line 5.7:** Please indicate the source of the test data.
- **Figure 2:** Please add a legend explaining the colors.
- **Figure 4:** Use either "correction" or "calibration" consistently throughout the text and figures.

---

## Author Comment (AC1)

1.General Comments:

Jin et al. clearly define the issue at hand and how they have contributed to the solution and what the resulting benefits are to science and society; namely, that common biomass burning emission inventories often omit diurnal information and vertical injection heights of fires, and that by incorporating their ideas, they have an emissions product for Southeast and East Asia (SEAF) that includes these two important pieces of information that ultimately improves the accuracy of models used for reporting and assessment of air quality, climate and public policy. They have done careful work to compare their results with measurements from TROPOMI, MISR and CALIPSO, and they present a comparison of their emissions dataset with common established BB emission datasets.

Of particular note is that they have been able to generate SEAF in such a way as to closely match not only the 2D structure of emissions observed from satellite, but the 3D structure as well, with the help of machine learning, albeit difficult for them to capture short-lived fires and emissions from low-lofted plumes. This is an encouraging contribution to the biomass burning emissions research community, and furthers the community's desire to see uncertainty in regional and global emissions decrease significantly. The main concern of this reviewer involves their method for generating and applying the diurnal FRP cycle. The authors need to include more discussion and review of the community's efforts regarding this very needed step in emission inventory development, and to show how their efforts are either similar to these previously published methods or are an improvement upon them.

Overall, I recommend this article for publication with minor revisions.

Response: We sincerely thank the reviewer for the comprehensive summary and the encouraging assessment of our work. We fully agree that the manuscript should better position our diurnal FRP-cycle method within the broader community efforts and more explicitly state what is inherited from prior approaches and what is new in our framework. We have revised the manuscript accordingly. **Please note that text in "italics and underlining" represents the revised sentences in the modified manuscript.**

**1. Enhanced review of community efforts on diurnal FRP modeling:** In the revised Introduction, we expanded the background discussion to provide a more complete and more balanced overview of the development of diurnal FRP modeling (Introduction, Lines 112–122: *A critical prerequisite for reducing uncertainty in emission estimates is the accurate reconstruction of the FRP diurnal cycle. Vermote et al., (2009) and Ellicott et al., (2009) established the foundational theoretical basis for using Gaussian functions to fit diurnal fire cycles from geostationary data. Subsequently, Kaiser et al., (2012) incorporated Kalman filtering in the Global Fire Assimilation System to address cloud gaps, while Andela et al., (2015) demonstrated the strong dependence of diurnal cycles on ecosystem types. More recently, Li et al., (2019, 2022) reconstructed sub-daily FRP variability by combining polar-orbiting and geostationary observations, with temporal gaps filled by integrating both available observations and ecosystem-specific diurnal climatologies, whereas Zheng et al., (2021) utilized Himawari-8 observations*

*to implement an event-based Gaussian representation of the FRP diurnal cycle, establishing a critical*

*regional reference for geostationary-based fire monitoring in East Asia.*).

**2. Clarification of similarities and improvements:** In response to your request to show how our efforts are either similar to, or an improvement upon, previously published methods, we now clearly distinguish between methodological heritage and innovation.

**(1) Similarity (heritage)**: As clarified in the revised Introduction and Methodology, our approach builds upon the community-established Gaussian representation of diurnal FRP cycles, thereby maintaining physical consistency with prior work. Historical statistics are used to determine the shape parameters of the diurnal curve, consistent with earlier Gaussian-based frameworks (Eq. (8)).

$$FRP(t) = ae^{-\frac{(t-b)^2}{2c^2}} + d \qquad (8)$$

*Where  $a, b, c$  are shape parameters controlling the amplitude, peak time, and width of the diurnal FRP cycle, respectively, derived from historical statistics for specific land cover types and regions.*

**(2) Improvement (innovation)**: We explicitly articulate the limitations of commonly used static climatological diurnal profiles, which are often applied when observations are missing and may introduce biases during non-active or extreme fire periods (Introduction, Lines 122–131: *Despite these advances, when observations are partially missing due to cloud occlusion or limited temporal sampling, many FRP-based emission frameworks still rely on static diurnal representations to fill gaps, such as superimposing predefined Gaussian-shaped curves or adopting climatological diurnal profiles that are invariant in time (Wooster et al., 2021). Such static treatments can lead to biases in emissions during non-active periods or dampen peak fire activity during extreme events, thereby introducing additional uncertainty into emission estimates. These limitations indicate that, although Gaussian-based representations of diurnal FRP cycles are widely adopted and physically grounded, their application in regions with frequent cloud cover and episodic extreme fires requires dynamic, event-specific adjustment rather than reliance on climatological averages.*). **To overcome these limitations, we introduce a two-stage adaptive fitting strategy (Eq. (8)).** *Where  $a, b, c$  are shape parameters controlling the amplitude, peak time, and width of the diurnal FRP cycle, respectively, derived from historical statistics for specific land cover types and regions. The parameter  $d$  is an event-specific additive adjustment that shifts the diurnal curve vertically to match the observed FRP level of a given day and can either increase or decrease the overall magnitude relative to the climatological baseline. Distinct from traditional static climatological Gaussian schemes,*

*we implement a two-stage adaptive fitting strategy. When sufficient instantaneous observations are available for a fire event (≥ 4 detections) and the reconstruction is stable ($R^2 ≥ 0.8$, indicating that the diurnal shape is sufficiently constrained by observations), all parameters ($a, b, c, d$) are jointly optimized to capture event-specific diurnal variability. However, when observations are sparse (< 4) or the reconstruction is unstable ($R^2 < 0.8$), the shape parameters ($a, b, c$) are fixed to their climatological values and only $d$ is solved from the available observations. This constrained formulation preserves a physically realistic diurnal shape while anchoring the overall magnitude to satellite observations, thereby reducing systematic mean-level offsets and, in a least-squares sense, providing a closer fit to the available observations than the pure climatological Gaussian baseline. Importantly, gap filling is not applied to all missing hours. Instead, the reconstruction is constrained to the high probability sustained burning windows determined in Table 1 for each region and vegetation type, and missing FRP is filled only within these statistically supported windows using Eq. (8) and the two-stage adaptive fitting strategy. This constraint prevents spurious reconstruction across low probability hours when sustained burning is unlikely, thereby preserving the physical realism of the reconstructed diurnal evolution.*

**(3) Evidence of improvement**: The effectiveness of the proposed dynamic adjustment is demonstrated by its dual capability to mitigate systematic underestimation while preventing artificial emission inflation.

- At the event scale, the method selectively recovers missing fire signals during critical windows. As stated in the revised manuscript: (Section 3.2.2: *Across all regions and vegetation types, reconstructed FRP profiles showed marked enhancements, particularly during key burning periods underrepresented in the original observations... In Region 1, cropland fires exhibited pronounced morning peaks (08:00–12:00 local time), with FRP increased by 67.7% after reconstruction... In Region 4, shrubland FRP increased by 88.7% during 16:00–00:00 local time, revealing active evening-to-night burning that was systematically underestimated due to twilight detection gaps.*). These localized adjustments ensure that missing fire activity is recovered based on physical evidence and agricultural practices.

- To ensure robustness when data are sparse, we implement a constrained logic: (Section 3.2.2: *In contrast, under observation-limited or unstable fitting conditions, the method employs a physically-constrained regularization strategy. By intentionally anchoring the diurnal shape to the established climatology and adjusting only the magnitude parameter, the framework effectively prevents the*

*introduction of artificial peak shifts or structural artifacts...*). This ensures we do not "over-fit" the data when observations are insufficient.

● At the regional scale, Fig. 7 demonstrates that the improvement is primarily a correction of systematic bias. We have added the following explanation: (Section 3.2.2: *The original observations systematically underestimate FRP due to cloud contamination, low temporal sampling frequency, and twilight detection blind zones... It should be noted that the relatively small changes in mean FRP compared to the standard deviation reflect systematic bias adjustment rather than random variability... The dynamically fitted FRP showed improved regional means (e.g., 23.85 MW in Region 2 and 17.49 MW in Region 4), representing relative increases of 3.52% and 1.62% compared to the original data.*).

● Overall, the dynamic adjustment framework improves the temporal representativeness of FRP by recovering underdetected fire activity during under-sampled periods, while avoiding artificial amplification associated with conventional climatological Gaussian fitting (Section 3.2.2: *Across all regions and vegetation types, reconstructed FRP profiles showed marked enhancements, particularly during key burning periods underrepresented in the original observations; Overall, the dynamically adjusted FRP profiles exhibited markedly improved temporal continuity compared with the original observations, capturing the primary peaks of fire activity and recovering missing signals during under-sampled periods; and the Gaussian fitting method ... resulting in systematic overestimation in certain regions, primarily due to artificial amplification during inactive periods*). As a result, the dynamically adjusted FRP yields more realistic regional mean values and improved statistical representativeness at the regional scale (Section 3.2.2: *The dynamically fitted FRP showed improved regional means ... representing relative increases of 3.52% and 1.62% compared to the original data; and the relatively small changes in mean FRP compared to the standard deviation reflect systematic bias adjustment rather than random variability*).

Specific Comments:

2.Lines 74-76: I believe this statement is a bit misleading. The authors are claiming here that the "integrated inventories" (i.e. the multi-source emissions datasets that incorporate diurnal cycles) generally have a large uncertainty when compared to the standard emission inventories. There may be two issues: 1) the authors are making the case that the standard inventories have uncertainty themselves and thus the need for integrated inventories, so although comparing against these standard inventories is important, we need to know if the areas where these standard inventories are weak and if the integrated inventories improve upon

them, and 2) Xu et al. 2023b is given as a paper that has assessed these integrated inventories and found large uncertainty (the numbers given in this sentence), but it seems as if the paper is only assessing their CHOBE inventory, which is not mentioned in their list of examples previously, and moreover, it does not assess any of the more well-known integrated inventories that the authors here present to us. It would be good to better present the issues with the integrated inventories that already exist, and to show how they improve upon the standard inventories but still underperform in areas that a new inventory can accomplish.

Response: Thank you for this careful and insightful comment. We fully agree that the statement in the original version of the manuscript regarding the uncertainty of "integrated inventories" was potentially misleading and did not clearly distinguish between the limitations of standard inventories and the advances achieved by multi-source integrated approaches. In response, we have substantially revised and restructured the Introduction to address this issue more clearly and transparently.

(1) We removed the generalized claim that integrated inventories exhibit "large uncertainty" and revised the associated citation strategy, avoiding reliance on Xu et al. (2023b), which evaluates a specific inventory rather than the broader class of integrated products. This revision ensures that the discussion of previous work remains contextually accurate and avoids inappropriate generalization.

(2) The revised Introduction now explicitly acknowledges the well-established improvements provided by existing integrated inventories. As stated in the revised text, recent studies combining geostationary and polar-orbiting FRP observations (e.g., Li et al., 2019,2022; Xu et al., 2023) have successfully produced hourly, kilometer-scale emission products that significantly improve spatiotemporal characterization compared with conventional daily or monthly inventories. This revision makes clear that integrated inventories represent a substantial methodological advancement over standard products.

(3) Rather than attributing remaining discrepancies to vague "uncertainty," the revised text now identifies a specific and common methodological limitation shared by many existing frameworks. We explain that when observations are partially missing due to cloud occlusion or limited temporal sampling, many FRP-based emission frameworks still rely on static or climatological diurnal representations for gap filling. Such static treatments can introduce systematic biases by inflating emissions during inactive periods or damping peak fire activity

during extreme events, particularly in regions with frequent cloud cover and episodic fires such as Southeast Asia.

(4) We have emphasized that while previous integrated inventories improved surface (2D) temporal characterization, they often lack the explicit, observation-driven vertical injection heights required for comprehensive 3D atmospheric modeling. This justifies the necessity of the new SEAF inventory, which provides an observation-driven, hourly 3D BB emission dataset for the broader SEA region by integrating a dynamic diurnal adjustment with a machine-learning-based vertical injection model.

By restructuring the Introduction in this way, we clarify that the motivation for SEAF is not a general deficiency of integrated inventories, but rather a remaining methodological gap in dynamic, event-specific reconstruction of diurnal FRP cycles and their consistent extension to three-dimensional emission representations.

3.Lines 76-79: It is mentioned that using a Gaussian scheme to reconstruct the diurnal cycle is a problem, but the authors use a Gaussian scheme themselves, so it should be specified better how the authors use the Gaussian scheme differently than the other integrated inventories. These works should be mentioned and discussed: Ellicott et al. 2009, Vermote et al. 2009, Andela et al. 2015, Kaiser et al. 2012, Li et al. 2019/2022, and Zheng 2021. Ellicott and Kaiser are not mentioned in the paper at all. Kaiser in particular provides an alternative method for cloud correction using a Kalman filter temporal prediction.

Response: We sincerely thank the reviewer for highlighting these foundational studies and for pointing out that our original manuscript did not sufficiently clarify how our Gaussian-based approach differs from previous integrated inventories. We agree that this distinction needed to be made explicit, and we have revised both the Introduction and Methodology accordingly.

**(1) Expanded and updated literature review of Gaussian-based approaches:** In the revised Introduction, we have substantially expanded the discussion of prior work to trace the development of diurnal FRP modeling within the community (see our response to General Comments (Item 1, above): **Enhanced review of community efforts on diurnal FRP modeling**)).

**(2) Clarification of how our Gaussian implementation differs from previous approaches:** As noted by the reviewer, our method also adopts a Gaussian representation of the diurnal FRP

cycle. However, the key distinction lies not in the functional form itself, but in how it is applied under observation-limited conditions. To avoid repetition, we refer the reader to our response to the previous Specific Comment (see our response to General Comments (Item 2, above): **Clarification of similarities and improvements**), where we detail the adaptive two-stage fitting strategy implemented in SEAF.

4.Lines 208-209: Using a beta value of 0.95 seems really high to me. Does this mean that if you have only 5% non-cloud, you still estimate the undetected FRP for the 95% cloud-filled portion of a grid cell from the 5%? Have you tried a lower threshold to see how the results change, or not?

Response: We thank the reviewer for raising this important concern regarding the choice of the cloud-fraction threshold. When the cloud fraction $\beta$ approaches 0.95, the cloud-correction scheme estimates grid-cell–aggregated FRP using the available clear-sky observations; however, this procedure is not a simple linear extrapolation from a small non-cloud fraction, nor does it assume spatial homogeneity within the grid cell.

**(1) Mathematical constraint by damping term**

The formulation in Eq. (2) explicitly includes a quadratic damping term ($\alpha = 0.25$) to constrain amplification under high cloud-cover conditions. This term suppresses the rapid growth that would otherwise occur under a simple reciprocal correction. For example, at $\beta = 0.95$, the effective correction factor is limited to approximately 3–4 times the observed clear-sky FRP, rather than the ~20-fold amplification implied by a purely linear extrapolation ($1/(1-\beta)$). This design ensures that the cloud correction remains physically bounded even under heavily clouded conditions.

**(2) Sensitivity analysis across cloud-fraction thresholds**

To evaluate whether a high cloud-fraction threshold introduces instability or systematic overestimation, we conducted a sensitivity analysis in which $\beta$ was varied from 0.70 to 0.95 for both NOAA-20 and Suomi-NPP VIIRS observations. The results show that total annual FRP estimates are highly stable across this range, with variations remaining below 0.5% (Fig. S2). This indicates that the cloud-correction scheme does not introduce numerical artifacts or bias regional-scale FRP budgets, even when a high threshold is applied.

**(3) Rationale for retaining $\beta = 0.95$**

Based on the combined mathematical constraint and the demonstrated empirical stability, a threshold of β = 0.95 was retained to avoid systematic underestimation of FRP in persistently cloudy regions of Southeast Asia, while maintaining numerical stability. This choice represents a balance between recovering cloud-obscured fire signals and preventing excessive amplification under extreme cloud-cover conditions.

A summary of this sensitivity analysis has been added to Section 2.3.1 of the revised manuscript (Lines 314–316), and the full results are provided in the Supplementary Material (Fig. S2).

5.Eqs. 3 & 4: If you combine the equations, the 1 cancels out and you are simply left with multiplying the AHI FRP by the ratio of VIIRS to AHI FRP for the coincident data. Perhaps it would be simpler to define r as simply the fraction, unless it is beneficial to center the ratios around zero.

Response: We thank the reviewer for this sharp observation regarding the algebraic formulation. We fully agree that for a single instantaneous calibration, substituting Eq. (4) into Eq. (3) is algebraically equivalent to simply scaling AHI FRP by the VIIRS-to-AHI ratio. However, as the reviewer correctly surmised, we deliberately defined $r_i$ in the form of a zero-centered relative bias $(1 + r_i)$ rather than a simple ratio, because this formulation offers significant advantages for statistical aggregation and numerical stability when calibration factors must be averaged across time and land-cover classes. We have revised Section 2.3.2 to explicitly clarify the statistical rationale behind this choice.

**(1) Statistical Robustness in Aggregation**: Unlike a simple instantaneous correction, our method requires aggregating these factors to generate daily ($r_{d,j}$, Eq. 5) and monthly calibration factors when simultaneous observations are unavailable. Averaging zero-centered relative bias terms is generally more robust and statistically consistent than averaging direct ratios, which can be asymmetric and sensitive to outliers.

**(2) Interpretability:** Defining $r_i$ as a relative bias allows $r_i = 0$ to serve as a direct, intuitive baseline for perfect consistency between sensors. Positive and negative values immediately indicate the direction of the bias (underestimation or overestimation).

**(3) Unified Calibration Framework:** This formulation provides a unified mathematical structure that connects the instantaneous bias (Eq. 4) with the dynamic daily aggregation (Eq. 5) and monthly land-cover-specific adjustments (Eq. 6). This hierarchy ensures that the

calibration logic remains consistent whether we are using a direct match or a temporal/spatial

fallback, maintaining the integrity of the emission inventory across all scales.

- 6.Eq. 6: I would be very interested to know if using ri instead of rml makes much of a difference in general. My idea would have been to prioritize the general calibration (rml) over of instantaneous calibrations (ri) in developing the diurnal cycle, but here you prioritize the instantaneous calibrations.

Response: We thank the reviewer for raising this important question regarding the calibration

hierarchy in Eq. (6). Below we clarify the distinct roles of the instantaneous calibration factor $r_i$

and the monthly land-cover-specific factor $r_{m,l}$ and explain why instantaneous calibration is

prioritized when available.

**(1) Separation between magnitude calibration and diurnal shape reconstruction:** Both $r_i$

and $r_{m,l}$ are applied solely to correct the magnitude (amplitude) of the AHI FRP to match the

polar-orbiting reference. They do not determine the shape (e.g., peak time and width) of the

diurnal cycle. As described in the Methodology, the shape parameters are derived from longterm AHI statistics (2016–2023) using Gaussian fitting, ensuring the structural stability of the

diurnal curve regardless of which calibration factor is used.

**(2) Rationale for Prioritizing Instantaneous Calibration ( $r_i$ ):** We prioritize the

instantaneous calibration factor ($r_i$) whenever a reliable simultaneous VIIRS-AHI pair ($\pm 5$ min)

is available. The primary reason is event-specific fidelity.

- Capturing Real Intensity: Using $r_i$ ensures that for a specific fire event, the AHI FRP

  magnitude is directly constrained by the high-quality VIIRS observation at that moment.

- Avoiding Over-smoothing: Prioritizing the general monthly average ( $r_{m,l}$ ) would

  artificially smooth out the variations of individual fire events. For example, during extreme

  fire episodes where intensities significantly exceed the climatological mean, relying on

  $r_{m,l}$ would systematically underestimate the peak energy.

**(3) The Adaptive Hierarchy:** Our approach functions as an adaptive hierarchy: instantaneous

calibration ($r_i$) is used to anchor event-specific magnitudes when available, while statistically

robust daily or monthly averages ($r_{m,l}$) are used as a fallback under data-sparse conditions. This

design maximizes physical fidelity at the event scale without sacrificing stability at longer

temporal scales. While differences between using $r_i$ and $r_{m,l}$ are most evident for individual

high-intensity events, their influence is largely smoothed at monthly or regional scales, supporting the use of this hierarchical strategy.

We have clarified this physical and statistical distinction in the revised Methodology to avoid any ambiguity regarding the impact of $r_i$ on the diurnal cycle structure.

- 7.Eq. 7: Did you mean to say "VIIRS/AHI FRP = 0" instead of "… < 0"?

Response: Yes. We thank the reviewer for pointing out this ambiguity. FRP is a physical quantity and cannot be negative. In the revised manuscript, we have reformulated Eq. (7) to remove the use of negative values and to explicitly distinguish between valid observations and missing data.

$$FRP_i^{fuse} = \begin{cases} FRP_i^{VIIRS} & i \in \Omega_{VIIRS} \\ \overline{FRP_i^{AHI}} & i \notin \Omega_{VIIRS} \ \wedge \ i \in \Omega_{AHI} \\ 0 & otherwise \end{cases} \tag{7}$$

- 8.Line 263: I may not be understanding correctly, but if I do, I think the term "T_gap" conveys the opposite of what is being described. "T_gap" makes me think it is the period between high intensity burn periods, not how long the high intensity burn lasts.

Response: We thank the reviewer for this helpful clarification. We agree that the notation "T_gap" was potentially misleading, as it could be interpreted as the interval between burning periods, whereas our analysis focuses on the duration of persistent burning.

To remove this ambiguity, we have revised the manuscript by renaming "T_gap" to "T_SB" (sustained burning), which explicitly denotes the maximum duration of sustained burning derived from burning frequency statistics. We also revised the accompanying text and Table 1 to clearly distinguish between the duration of sustained burning and the separation interval between adjacent burning windows. These changes clarify the physical meaning of the parameter and ensure consistent interpretation throughout the manuscript.

- 9.Lines 268-269: Please explain how these Gaussian curves were constructed, e.g. were the peaks adjusted before averaging between days to account for any daily differences?

Response: Thank you for this important clarification request. We agree that the construction of the Gaussian diurnal curves requires a clearer description, particularly regarding how daily variability is treated.

(1) As clarified in the revised manuscript, the climatological Gaussian diurnal curves are derived from long-term Himawari-8/9 FRP observations (2016–2023), grouped by region and vegetation type. For each local-time bin, FRP observations from the full 2016–2023 record are

aggregated to compute a multi-year mean FRP, yielding a 24-hour climatological mean diurnal profile. A Gaussian function is then fitted to this mean profile to estimate the characteristic diurnal shape (peak timing and width). No explicit peak shifting, temporal alignment, or normalization at the day level is applied prior to this fitting. The resulting curves are intended to represent statistically dominant diurnal fire behavior rather than day-specific variability (Lines 281-285: *Long-term AHI observations (2016–2023) are used to derive climatological Gaussian representations of FRP diurnal cycles (constructed by computing multi-year mean FRP at each local-time bin to form a 24-hour mean diurnal profile*).

(2) Treatment of daily variability: Daily differences in fire intensity or peak strength are not handled through pre-alignment or averaging of daily peaks. Instead, as detailed in our response to the General Comments (see our response to General Comments (Item 2, above): **Clarification of similarities and improvements**), daily variability is captured through a subsequent dynamic adjustment step. When sufficient same-day observations are available, an event-specific magnitude parameter is solved while the diurnal shape remains physically constrained by the climatological Gaussian form (Sections 3.2.2–3.2.3).

This clarification has been added to the revised manuscript to explicitly state that peak positions are not adjusted prior to averaging and that daily variability is accounted for through event-specific dynamic adjustment rather than temporal alignment of peaks.

- 10.Lines 340-342: citation needed

Response: Thank you for this comment. The relevant text (*Furthermore, comparisons of thermal anomaly observations from drones with both VIIRS and Himawari-8 data indicate that VIIRS measurements are more reliable.*) has been fully revised in the updated manuscript (Section 3.1). The previous qualitative statement has been removed and replaced with a physically grounded, quantitative discussion of cloud-induced FRP underestimation and cross-sensor differences, explicitly supported by established FRP-based emission and uncertainty studies (e.g., Kaiser et al., 2012; Andela et al., 2015b; Freeborn et al., 2014; Deng et al., 2024; Wickramasinghe et al., 2018; Hall et al., 2023). The revised section now includes region-specific analyses showing 2–9% increases in VIIRS FRP due to cloud correction and explains the physical mechanisms behind observed sensor differences. This ensures that all statements are now properly referenced and methodologically justified.

-  There doesn't seem to be any citations or analysis done to show how the increased estimates of FRP in cloud-filled regions compare to reality. Clearly, there must be an increase, but without any general idea of how much is missing due to cloud cover, it doesn't seem to decrease the uncertainty in the results.

Response: We thank the reviewer for this insightful comment. We agree that it is critical to demonstrate that the cloud-correction scheme provides a physically constrained improvement rather than an arbitrary increase in FRP estimates. Accordingly, we have revised Section 3.1 to clarify the robustness of our results from three complementary aspects.

(1) Physical basis and literature support: We explicitly document that persistent cloud cover leads to systematic underestimation of satellite-derived FRP, a limitation that has been widely recognized in FRP-based emission inventories and FRP field reconstruction studies (Kaiser et al., 2012; Andela et al., 2015b). The adopted coverage-based cloud-correction approach follows established practices in the literature and is therefore physically motivated rather than ad hoc.

(2) Conservative magnitude constrained by known uncertainties: We quantify that cloud correction leads to a modest increase of approximately 2%–9% in VIIRS-derived FRP across all regions. Importantly, this adjustment range is comparable to, or smaller than, inherent sensor-level FRP uncertainties (approximately 15–30%) and atmospheric attenuation effects reported in previous validation studies (Freeborn et al., 2014; Deng et al., 2024). This indicates that the correction is conservative and primarily reduces cloud-induced systematic low bias, rather than inflating FRP estimates beyond plausible physical bounds. The fact that the applied corrections fall well within the known sensor uncertainty envelope provides indirect validation of their physical reasonableness.

(3) Cross-sensor consistency as internal validation: Crucially, the framework does not uniformly increase FRP values. While cloud correction increases VIIRS FRP, the subsequent cross-sensor calibration results in an overall decrease in AHI FRP (approximately −3% to −24%) by mitigating sensor-specific overestimations related to mixed-pixel effects and false detections (Hall et al., 2023; Wickramasinghe et al., 2018). This bidirectional adjustment—with VIIRS increasing and AHI decreasing—demonstrates that the method targets systematic biases in both directions and improves the overall physical consistency of the resulting FRP fields. This bidirectional behavior provides an internal plausibility check on the correction magnitude and

demonstrates that the framework is not a uniform inflation scheme but a bias-correction system anchored to the cloud-corrected VIIRS reference.

These clarifications and the corresponding quantitative analyses have been incorporated into the revised Results section (Section 3.1). While direct ground-truth validation in persistently cloudy regions remains challenging, as acknowledged in the broader FRP validation literature, the combination of a physically grounded methodology, conservative adjustment magnitudes within known uncertainty bounds, and improved cross-sensor consistency indicates that the applied corrections are unlikely to amplify uncertainty and instead act to reduce systematic low bias in the FRP estimates.

12.Line 370 / Fig. S4: Please list the units for time in the caption – I can't seem to make the "9:00-16:00 local time" for NE China (region 4) cropland (column 1) correspond to the figure. The x-axis shows this to peak at ~6:00, which if in UTC would correspond to a local CST time of 14:00, which is not the center of 9:00-16:00.   Also, it seems as if for Region 1 the Gaussian doesn't work too well because of the prolonged right tail – perhaps a skew term should be introduced.

Response: We thank the reviewer for pointing out the inconsistency between the reported peak times and the time axis in Fig. S5 (the pre-revision Fig. S4). We agree that this required clarification.

(1) Time reference and unit clarification: In the revised manuscript, we have clarified that the x-axis in Fig. S5 represents hours in UTC. Correspondingly, all diurnal peak times reported in Section 3.2.1 have been revised to UTC, with local time provided only as a reference where necessary. For example, cropland fires in Northeast China (Region 4) peak between approximately 01:00 and 08:00 UTC, corresponding to 09:00–16:00 local time (UTC+8) (Section 3.2.1, Lines 591–592). These revisions ensure consistency between the figure and the text and remove ambiguity regarding the time reference.

(2) We agree that this asymmetry indicates a departure from an ideal symmetric Gaussian shape and merits explicit discussion. In the revised Discussion, we now clarify that the pronounced right tail in Region 1 is likely driven by region-specific fire and observational characteristics rather than by a deficiency of the Gaussian framework itself (*The prolonged right-tailed diurnal patterns observed in Region 1, which are potentially driven by the high thermal inertia of*

*smoldering peatlands or persistent late-afternoon cloud interference…*). Such conditions are common in tropical peatland-dominated regions and can lead to sustained or delayed fire radiative signals. We further acknowledge that, under these conditions, a symmetric Gaussian assumption may lead to a temporal shift or partial underestimation of evening emissions (*…a symmetric Gaussian assumption may lead to a temporal shift or underestimation of evening emissions*). Similar deviations from unimodal symmetric diurnal behavior have been reported in previous studies for tropical fire regions characterized by frequent cloud cover and long-lasting combustion processes (Page et al., 2009; Rein, 2009).

While a skewed fitting function could potentially better represent such asymmetric structures, we retain the Gaussian formulation in this study as a deliberate trade-off between computational robustness and the primary objective of climatological diurnal fitting, namely, the stable characterization of dominant fire peak timing for gap filling (*Although this simplification represents a necessary trade-off between computational efficiency and capturing primary fire peaks…*). This methodological choice is therefore intentional rather than an oversight, and we explicitly acknowledge this limitation and its associated uncertainty in the Discussion.

- 13.Lines 378-380 / Fig. 5: Would you please briefly discuss what is going on in panels g and l. How are the filled data points so far off the GLS in g, and why was the peak not shifted right in l?  Of all the panels, only a, m, p, q and t seem convincing and substantial; the rest of the panels have peaks that are difficult to corroborate with the presented data.

Response: Thank you for the reviewer's insightful observation regarding Fig. 5. We acknowledge that the previous presentation, which included unstable GLS fits with $R^2 < 0.8$ for demonstration, could be confusing. In the revised manuscript, we regenerated Fig. 5 to strictly follow our two-stage gap filling strategy. Each panel now displays only the fitting mode that is actually used for that event.

In the revised Fig. 5, panels a, p, q, s, and t illustrate cases where the within-day sampling is sufficient to constrain the diurnal curve shape ($R^2 \geq 0.8$, and $\geq 4$ detections), therefore the diurnal cycle is reconstructed using a least-squares GLS fitting. In contrast, the remaining panels, including g and l, represent cases with sparse or uneven observations where the diurnal shape parameters cannot be reliably constrained. In these cases, the framework switches to the GVM mode, in which the diurnal shape is fixed to region- and land-cover-specific

climatological parameters and only an event-specific additive term d is solved from the available detections.

This design directly explains the two reviewer questions. First, in panel g, the filled points are not expected to lie on the GLS curve because GLS is not applied in this information-limited case. Instead, the GVM reconstruction prioritizes physical plausibility and robustness by avoiding underdetermined shape optimization. Second, in panel l, the peak timing is not shifted to the right because the available observations do not provide sufficient constraints to justify a data-supported peak-time adjustment. Allowing a peak shift under such sampling would improve the visual fit locally but would introduce spurious timing changes that cannot be verified.

Although GVM retains the climatological diurnal shape, it is still mathematically and practically superior to using a pure climatological Gaussian curve without event-level adjustment. Specifically, the additive term d is solved in a least-squares sense to minimize residuals to the available observations, thereby correcting systematic mean-level offsets between the climatological baseline and the observations for that specific day. As a result, the GVM mode anchors the reconstructed magnitude to the observed FRP level while avoiding artificial peak shifts or peak broadening under sparse sampling conditions.

Overall, Fig. 5 is intended to demonstrate that the proposed two-stage strategy does not apply a single fitting approach to all cases. Instead, it uses GLS when the data support event-resolved fitting and switches to GVM when the sampling is insufficient, which is a conservative choice to prevent overfitting while maintaining physical consistency. This design choice is further supported by independent evaluations presented later in the manuscript, including improved peak timing statistics (Fig. 6), enhanced recovery of daily FRP totals (Fig. 7), and strong agreement between emission-derived CO and satellite observations.

- 14.Lines 419-440 / Fig. 7: The differences between the original and filled mean FRP values are so low compared with their standard deviations that it's hard to argue for the significance of these changes. Please address this. Also, do I understand correctly that the conventional Gaussian fits result in lower FRP for most of the regions? How did you define the conventional fit? If simply by not using GLS and GVM, then would you mind mentioning how e.g. Ellicott et al. 2009, Zheng et al. 2021, and Andela et al. 2015 all have dynamic diurnal Gaussian fits that change amplitude and/or duration with each day's data, and how you are similar or different in your approach.

Response: We thank the reviewer for this insightful comment. We address below the interpretation of the mean FRP differences in Fig. 7, the definition of the conventional Gaussian fit, and the relationship between our approach and previously published dynamic diurnal Gaussian methods.

(1) Interpretation of small mean differences relative to standard deviations: The relatively small changes in mean FRP compared with the large standard deviations shown in Fig. 7 do not indicate weak correction. Rather, they reflect the conservative nature of the reconstruction and the fact that the standard deviation primarily represents intrinsic spatiotemporal fire variability across heterogeneous fire regimes, rather than reconstruction uncertainty. In other words, the standard deviation shown in Fig. 7 is dominated by real fire intermittency and regional heterogeneity, not by the magnitude of the applied correction. Fig. 7 illustrates two opposing systematic effects: missing observations during sustained high-burning periods lead to systematic underestimation in the unfilled product, while unconditional application of diurnal Gaussian curves over a full 24-hour cycle tends to introduce artificial FRP during low- or non-burning periods. The modest net change in the mean therefore reflects bias redistribution, whereby underestimation during active burning windows is reduced while overestimation during inactive hours is suppressed. As a result, statistical significance should not be assessed solely by the ratio of mean change to standard deviation, because the correction targets systematic temporal bias rather than overall variance reduction.

(2) Definition and behavior of the conventional Gaussian fit: In this study, the "conventional Gaussian fit" refers to a static climatological diurnal curve derived from long-term averages and applied uniformly across the entire 24-hour cycle without conditioning on event duration, high-burning probability windows, or observation availability. This definition intentionally mirrors the most common implementation of climatological Gaussian gap filling used in many FRP-based emission frameworks. Under this definition, the conventional fit can yield either higher or lower regional mean FRP depending on the balance between recovered emissions during active periods and artificially assigned emissions during inactive periods. In several regions, the latter effect dominates, resulting in lower or even negative net corrections relative to the dynamically adjusted reconstruction shown in Fig. 7. This explains why the conventional

Gaussian approach produces lower FRP estimates for most regions in Fig. 7, despite partially filling missing observations.

(3) Relation to previous dynamic Gaussian approaches: We acknowledge that previous studies, including Ellicott et al. (2009), Andela et al. (2015), and Zheng et al. (2021), implemented dynamic diurnal Gaussian reconstructions that allow amplitude and/or duration to vary with daily observations. Our approach is conceptually aligned with these studies in recognizing the need for event-specific adjustment. However, it differs in that dynamic parameter optimization is explicitly conditioned on observational sufficiency and reconstruction stability. When event-day observations are insufficient or the fit is unstable, the diurnal shape is intentionally constrained to climatology and only the magnitude is adjusted. This strategy avoids unsupported peak shifts and prevents artificial inflation of FRP outside physically meaningful burning periods.

Overall, the small net differences in mean FRP shown in Fig. 7 should be interpreted as evidence of conservative, physically constrained bias correction rather than weak adjustment. The proposed method improves the realism of daily FRP distributions by redistributing bias while preserving the intrinsic variability of fire activity. This design choice prioritizes physical plausibility and stability over maximizing mean differences, consistent with the objectives of emission gap filling.

15. Lines 556-557: Can you remind the reader to reference Figure 12a for this claim about the comparisons to MISR?

Response: We thank the reviewer for this suggestion. We have revised Line 863 to explicitly reference Fig. 12 (a), which presents the MISR-based plume height comparison supporting this conclusion.

- 16. Fig. 4: Please explain how the cloud-corrected AHI FRP data can be less than the non-cloud-corrected data.

Response: Thank you for raising this important point. We clarify that the "correction" applied to AHI FRP in this study is not a pure cloud-recovery procedure and therefore does not necessarily lead to a monotonic increase in FRP. To avoid any further ambiguity, we have updated the caption of Fig. 4 to explicitly distinguish between the cloud-gap correction applied to VIIRS and the cross-sensor calibration applied to AHI.

In the revised manuscript, cloud correction is applied exclusively to VIIRS FRP to compensate for cloud-induced under-detection, whereas AHI FRP is subsequently adjusted **through cross-sensor calibration** using the cloud-corrected VIIRS product as a radiometrically more reliable reference. This calibration is designed to improve physical consistency in FRP magnitude between sensors rather than to inflate AHI-derived values. The results shown in Fig. 4 and Section 3.1 demonstrate that after cross-sensor calibration using cloud-corrected VIIRS FRP as a reference, calibrated AHI FRP exhibits region-dependent responses. In most regions, AHI FRP decreases due to suppression of sampling-related overestimation, whereas in regions with substantial BB emissions (e.g., northern Laos), AHI FRP still increases as cloud correction reveals additional fire activity previously underestimated. Therefore, calibrated AHI FRP may either decrease or increase depending on the regional balance between cloud-induced underestimation and sampling-related overestimation already documented in the manuscript.

- 17.Figs. 4, 7 & S7b: The resolution of the images seems to be too coarse. There appears to be a gridded/stripped pattern in the images – it appears to be an artifact in the data (particularly in Fig. 7), but it could just be the poor image resolution.   Please update the resolution – I would like to be able to see more detail when I zoom in.   Please also confirm what the stripped pattern is if it is indeed an artifact.

Response: Thank you for your careful review and the opportunity to clarify our visualization approach. We have carefully examined Figs. 4, 7, and S7b regarding the "grid/striping" patterns. We would like to clarify that these patterns are not instrumental artifacts (such as VIIRS detector striping) or algorithmic errors. Instead, they represent the native resolution of our gridded product. The "blocky" appearance is a deliberate choice for the following reasons:

(1) Data Fidelity and Transparency: We used a pixel-wise rendering method (pcolormesh with no interpolation). This ensures that each pixel's color accurately represents the underlying aggregated FRP value. Spatial interpolation (smoothing) often creates a false sense of continuity, which can misrepresent the magnitude of fire intensity and artificially "smear" fire signals into non-fire areas.

(2) Product Characteristics: Since the data is aggregated into a regular $0.03°×0.03°$ grid, the boundaries between grid cells are inherently discrete. The "striping" the reviewer observed is the alignment of these discrete grid cells at the boundaries of fire plumes.

To facilitate detailed inspection during the review process, we have additionally provided ≥300 dpi versions of Figs. 4, 7, and S7b in the response to reviewers, allowing the grid structure and spatial gradients to be examined clearly when zoomed in. The figures included in the manuscript follow the journal's standard publication resolution, and no spatial smoothing or interpolation has been introduced in either case.

[Figure]

**Figure 4**

[Figure]

**Figure 7**

[Figure]

**Panel from Figure S8 (formerly Figure S7)**

- 18.Fig. 11: I think the images would be easier to interpret if you kept the SEAF panel as is and converted the rest of the panels to difference maps.

  Response: We thank the reviewer for this helpful suggestion. Fig. 10 (formerly Fig. 11) has been revised accordingly. The figure caption and the corresponding text in the manuscript have been updated to reflect this change (Section 3.3.2).

- 19.Fig. 14: It appears to me that the SEAF emissions in panel d is greater than that of panel e, but in panel f, it is reported as lower. Is this a mistake, or is there a dynamic that is not visually observable with panels d and e?

Response: Thank you for raising this point. We apologize that the distinction between panels (a–e) and panel (f) was not sufficiently clear in the original manuscript. The apparent inconsistency does not indicate an error, but rather reflects that these panels represent different quantities.

Panels (a–e) show the layer-resolved $PM_{2.5}$ emission mass after vertical allocation to five altitude layers (0.025 km, 0.275 km, 1.0 km, 2.75 km, and 5.5 km). In SEAF, smoke plume heights (SPH) are first predicted using our machine-learning model, and the predicted SPH is then used to guide the vertical allocation of emissions. Importantly, the allocation scheme distributes the emission mass across the layers that a plume is assumed to traverse, rather than assigning all mass exclusively to the highest layer. Consequently, the 2.75 km layer (panel d) can integrate contributions from both moderate injection cases and the lower portions of high-reaching plumes, whereas the 5.5 km layer (panel e) only contains the fraction assigned to the uppermost layer. This mechanism can lead to larger allocated emission mass at 2.75 km than at 5.5 km, even when high SPH occurrences are present. Panel (f), in contrast, compares the relative frequency distribution of SPH, that is, the frequency with which plume tops fall within each height interval. Therefore, the frequency at 5.5 km being higher than at 2.75 km does not imply larger emission mass allocated to the 5.5 km layer. As discussed in the revised text, the SEAF SPH distribution exhibits a relative dip near 2.75 km, which is consistent with a regime in which plumes either remain largely within the boundary layer or are lofted to higher altitudes under strong convection, resulting in fewer plume tops occurring at intermediate heights. Notably, SEAF retains a non-negligible occurrence at 5.5 km (approximately 0.12), which is broadly consistent with the MISR and CALIPSO distributions, while conventional inventories such as GFAS v1.2 tend to under-represent these high-altitude injection signals.

To prevent misinterpretation, we have revised panel (f) by changing the y-axis label from "Height" to "Smoke Plume Heights (SPH)", and we have updated the Figure 14 caption to explicitly distinguish layer-allocated emission mass in panels (a–e) from the SPH frequency distribution in panel (f).

- 20.Fig. S7: The color scale is a bit unhelpful since the values saturate too quickly to be able to do any useful visual comparisons. Either stretch the scale, or as suggested with Fig. 11, convert the images to difference maps. Also, the units need to be changed to do proper comparisons

since the spatial resolutions are different between the panels. Please convert them to Mg/yr/km^2.

Response: We thank the reviewer for this helpful suggestion. Fig. S7 has been revised by rescaling the color bar to avoid saturation and by converting all emission fields to area-normalized units (Mg yr$^{-1}$ km$^{-2}$), thereby enabling consistent spatial comparison across inventories with different native resolutions.

Technical Corrections:

- 21.Line 72: missing period

  Response: Thank you for pointing out this minor typographical issue. We note that the sentence referred to in the original manuscript has been removed in the revised version as part of a substantial reorganization and condensation of the Introduction section. The revised Introduction now presents a more concise and focused discussion of prior work on diurnal FRP reconstruction and multi-source fire emission inventories, and the specific punctuation issue at Line 72 is therefore no longer applicable.

- 22.Line 77: "understate" might not be the best word, maybe underestimate, limit, etc.

  Response: Thank you for this helpful suggestion regarding word choice. We note that the sentence containing the term "understate" in the original manuscript has been removed in the revised version as part of a substantial reorganization of the Introduction.

- 23.Line 120: missing space

  Response: Thank you for pointing this out. The missing space has been corrected in the revised manuscript.

- 24.Line 332: delete first comma?

  Response: Thank you for pointing this out. The formatting has been corrected to "Texas Commission on Environmental Quality (2022)" in the revised manuscript.

---

## Author Comment (AC2)

General Comments:

This paper generates three-dimensional biomass-burning emissions for Southeast and East Asia by developing new fire diurnal cycles and vertical injection profiles. The proposed diurnal cycle is derived by integrating fire radiative power data from both geostationary and polar-orbiting satellite observations. The vertical injection profile is produced using a machine-learning model trained on satellite-retrieved smoke plume heights and meteorological variables. However, the manuscript is not well-structured, and the methodology lacks clarity and robustness. Substantial revisions are needed before the work can be considered for publication.

Response: We thank the reviewer for this comprehensive assessment. We agree that the previous version of the manuscript did not sufficiently convey the methodological coherence and robustness of the proposed framework. We have conducted a substantial revision of the manuscript, focusing on restructuring, clarification, and strengthening of the methodological presentation, as summarized below. **Please note that text in "*italics and underlining*" represents the revised sentences in the modified manuscript.**

**(1) Structural reorganization and methodological integration.**

The manuscript has been reorganized to explicitly present the SEAF product as a single end-to-end workflow, rather than as independent derivations of diurnal emissions and plume heights. The revised structure emphasizes the logical sequence from FRP fusion and diurnal reconstruction to smoke plume height prediction and final 3D emission construction. This integrated framework is now explicitly summarized in Eq. (12) and visually illustrated in the revised Figure 2, clarifying that diurnal variability and vertical injection are coupled at consistent spatial and temporal resolutions.

**(2) Improved methodological clarity and robustness.**

To enhance transparency and reproducibility, we have expanded the description of key methodological components, particularly the machine-learning framework for SPH prediction. Detailed information on dataset composition, sample selection, training–testing strategy, hyperparameter tuning, and performance metrics has been added. In addition, a dedicated validation subsection (Section 2.3.6) has been introduced to systematically describe the evaluation of diurnal emissions, plume heights, and the resulting 3D emission structure using multiple independent observational datasets and statistical indicators.

**(3) Strengthened regional context and evaluation framework.**

The Introduction has been expanded to better reflect prior studies conducted in SEA, including work on agricultural burning, peatland fires, combined VIIRS–AHI emission estimation, and extreme regional fire events. These additions clarify how the present study builds upon existing regional research while addressing key gaps, particularly the lack of a consistent hourly 3D emission inventory for this region. Furthermore, the evaluation framework has been strengthened by incorporating additional widely used emission inventories (including GFED v5.1), providing clearer interpretation of inter-inventory differences, and explicitly justifying the use of satellite-based CO as an observation-based tracer for BB emission evaluation.

We are confident that these comprehensive revisions effectively address the reviewer's concerns regarding structure, clarity, and robustness, significantly enhancing the overall scientific quality of the manuscript.

**Major comments:**

**1. Weak integration between the fire diurnal cycle and vertical injection profile components**

Although the overarching objective is to develop a three-dimensional biomass-burning emission dataset, the manuscript presents the derivation of the fire diurnal cycle and the vertical injection profile as largely independent processes. The authors first generate and validate 2-D fire emissions, then separately develop and validate smoke plume heights. However, the connection between these two components—and how they integrate to form the final 3-D emissions—is not clearly articulated. I recommend extensively restructuring the manuscript to better emphasize the methodological coherence and the interdependencies between these two parts.

Response: We thank the referee for pointing out that the previous manuscript structure may have given the impression that the diurnal emission reconstruction and the vertical injection profile were treated as independent components. We agree that this was an issue of presentation rather than methodology.

To address this, we revised the manuscript to explicitly present the construction of the SEAF product as a single end-to-end workflow. The integrated generation of hourly 3D emissions is now summarized by **Eq. (12)**, which explicitly links fused hourly FRP, column-integrated emissions, Random Forest–predicted SPH, and layer-wise vertical allocation. This formulation clarifies that FRP simultaneously controls the temporal

evolution of column emissions and serves as an input to the plume height prediction, which constrains the vertical distribution. In addition, **Figure 2** has been reorganized to highlight the unified workflow from hourly FRP to final 3D emissions, emphasizing that diurnal variability and vertical injection are coupled at the same spatial and temporal resolution. The evaluation of diurnal emissions and plume heights is conducted separately as a modular validation strategy to isolate uncertainties from different process components, rather than implying methodological independence. In practice, both the diurnal reconstruction and SPH prediction are applied to the same hourly FRP fields, ensuring that temporal variability and vertical allocation are driven by a consistent set of event-scale fire intensities.

2. **Insufficient background and regional context**

The literature review does not adequately cover relevant studies conducted in Asia. Much of the discussion focuses on work from the United States or other regions, while several closely related studies in Asia are overlooked. These omissions weaken the contextual grounding of the study and obscure how this work builds upon or differs from existing regional research. The authors should expand the background section to include key Asian studies and clearly articulate their linkages to the current work.

References:

'Dynamics of fire plumes and smoke clouds associated with peat and deforestation fires in Indonesia'

'Fire Particulate Emissions from Combined VIIRS and AHI Data for Indonesia, 2015-2020' 'Improved estimation of fire particulate emissions using a combination of VIIRS and AHI data for Indonesia during 2015–2020'

'Highly anomalous fire emissions from the 2019–2020 Australian bushfires'

Response: We sincerely thank the reviewer for this constructive comment regarding the insufficient background and regional context. We acknowledge that the previous version of the manuscript did not adequately integrate key Asian studies, which are essential for properly grounding this work in the specific fire regimes of SEA. To address this concern, we have substantially revised and expanded the Introduction to explicitly incorporate the suggested regional literature and to clearly articulate how the present study builds upon, and extends beyond, existing Asian research. The main revisions are summarized as follows.

(1) Strengthening the description of regional fire regimes in SEA.

We have expanded the background discussion to explicitly highlight the unique fire characteristics of SEA, with particular emphasis on peatland and deforestation fires in Indonesia. These fires exhibit distinct smoke plume dynamics and emission potentials compared to other regions, with strong implications for regional radiative balance (Tosca et al., 2011). This revision is reflected in the Introduction, where we now state that (*Notably, the SEA region exhibits unique fire regimes that require dedicated regional focus. For instance, peatland and deforestation fires in Indonesia possess distinct smoke plume dynamics and emission potentials compared to other regions, exerting a strong influence on the regional radiative balance (Tosca et al., 2011).*)

(2) Incorporating extreme climate-driven fire events and associated uncertainties.

We have added discussion of how climate forcing under warming and drying conditions can amplify fire emissions and increase uncertainties in conventional emission inventories, with explicit reference to El Niño–related extreme fire events in SEA. In particular, the severe Indonesian fires in 2015 are now cited as a representative example (Field et al., 2016; Huijnen et al., 2016). This is reflected in the revised Introduction, where we note that (*Furthermore, climate forcing under warming and drying conditions can substantially amplify fire emissions and drive strong deviations from climatological means, thereby increasing uncertainties in conventional emission inventories (Li et al., 2021). This effect is particularly relevant in SEA, where extreme fire activity frequently occurs during El Niño-related droughts, such as the severe Indonesian fires in 2015 (Field et al., 2016; Huijnen et al., 2016).*).

(3) Clarifying the linkage to existing VIIRS–AHI–based Asian studies.

We have explicitly incorporated recent regional studies that combine VIIRS and AHI observations to improve high-frequency fire emission estimates in Asia, including work focused on SEA (Lu et al., 2022; Li et al., (2019, 2022); Zheng et al., (2021)). These studies are now discussed in the Introduction to acknowledge prior regional efforts in multi-sensor FRP fusion ((1) *Recent studies have attempted to combine VIIRS and AHI data to improve high-frequency particulate emission estimates in specific areas such as Indonesia (Lu et al., 2022);*(2) *More recently, Li et al., (2019, 2022) reconstructed sub-daily FRP variability by combining polar-orbiting and geostationary observations,*

*with temporal gaps filled by integrating both available observations and ecosystem-specific diurnal climatologies, whereas Zheng et al., (2021) utilized Himawari-8 observations to implement an event-based Gaussian representation of the FRP diurnal cycle, establishing a critical regional reference for geostationary-based fire monitoring in East Asia.* ).

(4) Explicitly identifying the remaining regional gap addressed by this study.

Building on the expanded regional background, we now explicitly clarify that several key methodological limitations remain in existing Asian biomass-burning studies, which are directly addressed in the present work. These limitations include (i) the reliance on static diurnal representations to reconstruct FRP under conditions of cloud occlusion or limited temporal sampling, which can bias emissions during non-active periods or dampen peak fire activity during extreme events, and (ii) the lack of plume-height-resolved emissions and the absence of a consistent hourly three-dimensional emission framework across SEA.

These points are now explicitly stated in the revised Introduction ((1) *Despite these advances, when observations are partially missing due to cloud occlusion or limited temporal sampling, many FRP-based emission frameworks still rely on static diurnal representations to fill gaps, such as superimposing predefined Gaussian-shaped curves or adopting climatological diurnal profiles that are invariant in time (Wooster et al., 2021). Such static treatments can lead to biases in emissions during non-active periods or dampen peak fire activity during extreme events, thereby introducing additional uncertainty into emission estimates. These limitations indicate that, although Gaussian-based representations of diurnal FRP cycles are widely adopted and physically grounded, their application in regions with frequent cloud cover and episodic extreme fires requires dynamic, event-specific adjustment rather than reliance on climatological averages*; (2) *these studies are generally limited to two-dimensional (2D) emission estimates and specific fire episodes or subregions, without explicitly resolving plume injection heights or providing a consistent hourly 3D emission framework across SEA*). By explicitly identifying these gaps, the revised Introduction now clearly

motivates the development of the SEAF inventory as a systematic, high-resolution, observation-driven hourly three-dimensional BB emission dataset for SEA.

**3. Methodological issues and lack of robustness**

**• Choice of VIIRS 375-m product:**

The manuscript uses 375-m VIIRS FRP, but this product is known to saturate under highintensity fires. Why was the 375-m product chosen instead of the 750-m VIIRS FRP? This choice needs stronger justification beyond the higher spatial resolution. Have the authors tested or compared the 750-m FRP?

Response: Thank you for the reviewer's professional inquiry regarding the applicability of the VIIRS 375 m active fire product. We agree with the reviewer that the VIIRS 375 m I-band, particularly the I4 channel, may be subject to detector dynamic range limitations under extremely high-intensity fire conditions, potentially leading to brightness temperature saturation or folding and thus affecting I-band-based fire detection and classification (Schroeder et al., 2020; Zhang et al., 2017).

**It is important to clarify that the VIIRS 375 m active fire product used in this study (e.g., JPSS VIIRS Products-VIIRS Active Fires I-Band EDR) does not estimate FRP directly from the I4 channel.** According to the NOAA VIIRS I-band active fire algorithm, the I-band observations are primarily used for precise fire pixel detection and classification, while the quantitative FRP retrieval is based on the co-located 750 m M13 dual-gain mid-infrared radiance data. In practice, the FRP retrieved for a single 750 m M13 pixel is distributed among the coincident 375 m I-band fire sub-pixels, resulting in an FRP product reported at 375 m spatial resolution (Schroeder et al., 2020).

Therefore, although the FRP is reported at 375 m resolution, its physical retrieval already incorporates information from the M-band, and it is not equivalent to an FRP estimate derived solely from I-band radiances. Previous studies have further demonstrated that, owing to its larger pixel size and substantially enhanced dynamic range, the M13 band is rarely affected by saturation under active fire conditions. For example, Zhang et al., (2017) showed that in agriculturally dominated regions, while some intense fires may saturate the I4 channel, the M13 band remained largely unsaturated and was able to provide stable and reliable FRP estimates at the locations of I-band-detected fire pixels.

Based on these product characteristics, we selected the VIIRS 375 m FRP product as the primary fire radiative input for this study. Fire activity in SEA is strongly influenced by agricultural burning and is characterized by highly fragmented spatial patterns, with a large number of small and edge fires, making fire detection particularly sensitive to spatial resolution and omission errors (Huang et al., 2024; Vadrevu et al., 2022; Yin, 2020). Compared with the 750 m active fire product, the VIIRS 375 m product provides more accurate fire localization and a better representation of spatial heterogeneity, which is critical for the construction of high-resolution emission inventories and for the analysis of transboundary pollutant transport (VIIRS I-Band 375 m Active Fire Data | NASA Earthdata, 2025; Schroeder et al., 2014).

To avoid any potential ambiguity, we have also added a brief clarification in Section 2.1.1 (*Although the product is reported at 375 m spatial resolution, the I-band observations are mainly used for fire detection and localization, whereas FRP is retrieved based on the co-located 750 m M13 dual-gain mid-infrared radiance data and then allocated to the detected 375 m fire pixels…*) of the revised manuscript, explicitly stating that although the VIIRS product is reported at 375 m resolution, FRP is retrieved using the co-located 750 m M13 dual-gain mid-infrared radiance data following the standard VIIRS I-band active fire algorithm.

4• **AHI FRP correction:**

The reference (Li et al., 2022) cited for the AHI FRP correction does not actually use AHI FRP, making this citation inappropriate.

Response: We agree that Li et al. (2022) does not involve Himawari-8/9 AHI FRP and have therefore removed this citation. The manuscript now cites Xu et al. (2022, 2023), which explicitly apply VIIRS-referenced calibration to correct AHI FRP ( *Following (Xu et al., 2022, 2023), cloud-corrected VIIRS FRP was therefore adopted as an external reference to perform cross-sensor calibration of Himawari-8/9 AHI FRP using collocated observations, with the aim of reducing systematic biases associated with sensor characteristics and spatial resolution differences*), and clarifies that the specific cloud-corrected, multi-level fallback calibration framework is developed in this study.

5Unclear machine-learning description:

The section describing the machine-learning method lacks clarity. Important details such as the training–testing strategy, sample selection, and dataset composition are not provided.

Response: Thank you for this constructive comment. We agree that providing more technical details of the machine-learning workflow is essential for reproducibility. Following the reviewer's suggestion, we have significantly expanded Section 2.3.5 to explicitly document the modeling procedure. The specific additions are summarized below.

(1) Dataset composition and sample selection:

We clarified the data cleaning and filtering procedure used to construct the training dataset. (*Raw satellite observations were first spatially filtered to match the study domain, and records with non-physical values or missing FRP information were removed. This procedure resulted in a finalized dataset comprising 2,127 samples.*)

(2) Training–testing strategy:

A standard random split was implemented to ensure model generalizability and independent evaluation. (*For model development, the dataset was randomly divided into a training set (80%) and an independent testing set (20%).*)

(3) Hyperparameter tuning:

To ensure model robustness and avoid overfitting, we applied a grid-search strategy combined with cross-validation. (*A grid search combined with 5-fold cross-validation was employed to optimize model hyperparameters, yielding an optimal configuration of 200 trees (n_estimators) with a maximum tree depth of 10.*)

(4) Model performance:

Quantitative performance metrics were added to demonstrate the predictive reliability of the final model. (*The finalized RF model demonstrated strong predictive skill, with a root mean squared error (RMSE) of 334.68 m and an $R^2$ of 0.90 on the test set.*)

6• **Missing validation subsection in the Methods:**

A dedicated validation subsection should be added to the Methods section to clearly explain the evaluation workflow.

Response: Thank you for this constructive suggestion. We agree with the reviewer and have added a dedicated validation subsection to the Methods section. Specifically, a new subsection entitled "**2.3.6 Validation and evaluation strategy**" has been included.

This subsection systematically describes the evaluation workflow, including the validation of Random Forest–predicted smoke plume heights using independent MISR observations with statistical metrics (RMSE, $R^2$, R, and bias), the assessment of two-dimensional emissions through comparisons with TROPOMI CO columns and multiple BB emission inventories, and the evaluation of the three-dimensional emission structure using MISR, CALIPSO, and existing injection-height schemes. Detailed validation results are presented in Section 3. The validation strategy follows the same hierarchical structure as the emission construction, progressing from diurnal FRP evaluation (2D), to plume height prediction, and finally to the integrated 3D emission structure.

7• **Lack of GFED products in evaluation:**

It is unclear why the widely used GFED-related products are omitted from the evaluation. Including them would provide a more comprehensive comparison.

Response: Thank you for this constructive suggestion. We agree that including GFED-related products provides a more comprehensive benchmark and strengthens the inter-inventory comparison. We have therefore incorporated GFED v5.1 into our evaluation framework and updated the corresponding figures accordingly.

(1) Reason for the initial omission and clarification of product availability:

In the initial version, GFED v5.1 was not included because it was not publicly available at the time of manuscript preparation, and the latest officially released emissions only extended to 2022, which did not cover the 2023 study period analyzed in this work. To ensure temporal consistency across all inter-comparisons, GFED was therefore not included in the initial evaluation. We have now utilized the recently released extended version of GFED v5.1 and incorporated it into the revised manuscript, avoiding any ambiguity regarding the product release cycle.

(2) Revisions implemented in the manuscript:

We have updated Figures 10–11 and Figures S6–7 to include comparisons with GFED v5.1. This addition enables a more complete comparison against a widely used global burned-area based inventory and complements the FRP-constrained products discussed in this study.

(3) Value added to the evaluation framework:

Including GFED v5.1 strengthens the evaluation by placing SEAF in the context of both FRP-based and burned-area-based global emission frameworks, thereby improving the interpretability of inter-inventory differences and further supporting the robustness assessment of the SEAF product.

8• **Section 3.3.2 — Rationale for comparison with inventories:**
The manuscript evaluates the new emissions against existing emission inventories but mainly describes the differences without explaining the underlying causes. For example, why are the results lower than FINN but closer to FEER and QFED? Additional interpretation is needed in the Results or Discussion sections.

Response: Thank you for this constructive comment. We agree that providing a physical and methodological rationale for the inter-inventory discrepancies is essential for demonstrating the robustness of the SEAF inventory. Following your suggestion, we have added detailed interpretations in both the Results (Section 3.3.2) and the Discussion (Section 4) to clarify why SEAF estimates are lower than FINN but closer to FEER and QFED. The revisions explain the underlying causes from the following aspects.

(1) Methodological framework (FRP-based vs. burned-area-based): We clarify that SEAF, FEER, QFED, and IS4FIRES are all constructed within a top-down framework constrained by fire radiative power (FRP), whereas FINN relies on a burned-area-based approach. (*By resolving heterogeneous emission structures and sharp spatial gradients, SEAF captures small-scale fire clusters and localized hotspots that are often omitted or attenuated in burned-area-based products like GFED v5.1, which tend to display comparatively smooth and spatially diffuse emission patterns.*)

(2) Dynamic temporal representation of fire activity: A key driver of the inter-inventory differences lies in how fire activity is characterized over time. We emphasize that SEAF explicitly reconstructs sub-daily fire variability through a dynamic diurnal adjustment. (*Supported by the dynamically reconstructed diurnal FRP patterns, SEAF reproduces a pronounced seasonal peak of approximately 500 Gg month$^{-1}$ in Region 2, whereas inventories relying on infrequent sampling from polar-orbiting sensors, such as GFAS v1.2 and FEER v1.0, tend to underestimate this seasonal maximum.*)

(3) Treatment of smoldering-dominated peatland fires: We further discuss peatland-dominated fires, which are prevalent in parts of Southeast Asia and remain challenging for FRP-derived top-down approaches because deep smoldering combustion may be weakly expressed in FRP observations. (*We acknowledge that FRP-constrained approaches, including SEAF, are generally more conservative in regions where smoldering combustion is prevalent because cool fires are more difficult to detect and quantify using thermal infrared sensors compared to burned-area algorithms, which remain highly sensitive to assumptions on fuel consumption and burning depth.*)

(4) Quantitative and spatial consistency with observations: These methodological differences are reflected in the quantitative results and spatial patterns. (*For 2023, SEAF's annual PM2.5 estimate of 2362 Gg yr-1 lies within the central range of the inter-inventory spread, showing close agreement with FEER v1.0 and QFED v2.6r1 while corresponding to a reduction of approximately 67% relative to the burned-area-driven FINN v2.5.1.*)

Note on emission factors: We emphasize that while emission factors are a known source of uncertainty in BB inventories, all inventories compared in this study apply emission factors derived from similar literature-based compilations. Accordingly, the observed spread among inventories over Southeast Asia arises primarily from differences in fire activity characterization, temporal representation, and spatial resolution, rather than from emission factor selection alone.

9**• Lines 321–324 — Transferability of methods:**
These lines describe a core component of the method, yet the supporting references are based on studies from the U.S. and Europe. The authors should justify whether such methodologies are appropriate for Asian fire regimes.

Response: Thank you for raising this important comment. We agree that some of the references supporting this methodological component were originally developed and validated under fire regimes in the United States and Europe, and that the transferability of these approaches to Asian fire conditions therefore requires clarification.

(1) We have revised the Introduction to explicitly clarify why methodologies developed in other regions remain applicable to SEA (**see our response to 2. Insufficient background and regional context**).

(2) In the revised manuscript, we now explicitly clarify that the core of the proposed methodology is grounded in general physical relationships between FRP, energy release, and biomass combustion (Eq. 12). These relationships are not region-specific and have been demonstrated to be broadly applicable across different fire types and fire regimes.

(3) More importantly, the present study does not directly transplant existing methods to the Asian region. Instead, the approach is specifically adapted through region-dependent calibration and constraints to better represent Asian fire conditions. These adaptations include the use of VIIRS and AHI observations over SEA, regionally optimized diurnal reconstruction schemes, and evaluation against independent observations (e.g., TROPOMI CO) as well as multiple existing emission inventories over the SEAF domain. The spatial patterns, seasonal evolution, and regional consistency shown in Figures 10–11 and Figure S8 provide empirical evidence for the robustness of the method under Asian fire regimes characterized by fragmented land cover, agricultural burning, and peatland fires.

10• **Line 465 — Limited validation case:**
Validating the method using only a single biomass-burning episode is insufficient. More cases are needed to demonstrate robustness.

Response: Thank you for this valuable comment. We agree that validation based on a single BB episode is insufficient to establish full robustness at the event scale. In the revised manuscript, we have clarified the role of the event-scale analysis, explicitly acknowledged this limitation, and explained how the overall robustness of the SEAF inventory is supported by broader evaluations. The revisions are summarized as follows:

(1) Clarification of scope and acknowledgment of limitations: We explicitly distinguish between physical plausibility at the event scale and methodological robustness at regional and longer temporal scales. (**4. Discussion** *In addition, validation based on a single BB episode is insufficient to establish full robustness at the event scale (Figure 9). In this study, the event-scale analysis serves to demonstrate the physical plausibility of the fused fire emission product, whereas methodological robustness is primarily supported by regional- and long-term statistical consistency across multiple emission inventories.*)

(2) Observational constraints in SEA: We provide a practical explanation for the limited availability of suitable event-scale validation cases in the study region. (**4. Discussion** *Furthermore, systematic inspection of operational satellite imagery (e.g., NOAA STAR, https://www.star.nesdis.noaa.gov/mapper/) reveals that the identification of isolated, cloud-free plumes in SEA is severely constrained by persistent cloud cover and plume interference. Therefore, additional observational evidence is required to further validate the method at the event scale*)

(3) Robustness supported by multi-scale evaluation:

We emphasize that the main conclusions of this study do not rely on the single-event analysis alone, but are supported by extensive regional and multi-temporal evaluations presented throughout Section 3, including comparisons with independent satellite observations and multiple emission inventories.

(4) Outlook for future validation: We note that additional event-scale validation will be pursued as more suitable observational constraints become available. (**4. Discussion** *Therefore, additional observational evidence is required to further validate the method at the event scale as data availability improves*.)

We believe that this transparent discussion of the limitations of event-scale validation, together with the comprehensive regional and statistical evaluations presented in the manuscript, adequately addresses the reviewer's concern.

**11.Errors in basic information**
There are several factual inaccuracies that need correction. For example:
• Mixing up sensor and satellite names (line 97).

Response: Thank you for pointing this out. The issue has been corrected in the revised manuscript.

12• Stating that MODIS fails to capture nighttime events (lines 131–132).

Response: Thank you for pointing this out. We have revised the text.

13• Citing a reference for AHI FRP correction that does not actually use AHI FRP.

Response: Thank you for pointing this out. Please see our response to **4• AHI FRP correction**.

**Minor to moderate comments:**
14• **Lines 33–34:** Please specify the versions of all datasets used—at minimum in the Data section—since different versions may produce substantially different values.

Response: Thank you for this comment. We agree that specifying dataset versions is essential for reproducibility and comparability. We have revised both the Abstract and the Data section accordingly. In the revised Abstract, all emission inventories included in the inter-comparison are now explicitly identified with their corresponding version numbers, as reflected in the following sentence: (_with estimates lower than FINN v2.5.1 (67%) and GFED v5.1 (25%), but closely aligned with FEER v1.0, QFED v2.6r1, and IS4FIRES v2.0._) In addition, the Data section has been updated, and Table S2 now comprehensively documents the versions of all datasets used in this study, ensuring transparency and consistency throughout the manuscript.

15• **Lines 33–34:** Why not provide comparison results for all emission inventories included in the study?

Response: We appreciate the reviewer's suggestion. We agree that the scope of the emission inventory comparison should be clearly communicated in the Abstract. In the revised Abstract, we now explicitly state that SEAF is compared against six widely used global BB emission inventories, and we summarize the key quantitative relationships in a concise and balanced manner, as shown in the following revised sentence: (_Annual $PM_{2.5}$ emissions in SEAF are approximately 2362 Gg $y^{-1}$, placing it within the central range of six widely used global BB inventories, with estimates lower than FINN v2.5.1 (67%) and GFED v5.1 (25%), but closely aligned with FEER v1.0, QFED v2.6r1, and IS4FIRES v2.0.)_ Detailed comparisons for all inventories are presented and discussed in the main text and supplementary material.

16• **Line 35:** Please use statistical metrics to demonstrate the performance of the SPH estimation rather than presenting a single numerical value.

Response: Thank you for this comment. We agree that a single numerical value is insufficient to characterize the performance of smoke plume height (SPH) estimation. In the revised Abstract, we now report multiple statistical metrics to quantitatively evaluate SPH prediction performance, as shown in the following revised sentence: _The RF-SHAP framework successfully predicts SPH ($R^2 = 0.90$, RMSE = 335 m) with over 90% of estimates within ± 500 m._

17• **Line 38:** "Satellite observations" should be specified. If you mean MISR SPH, please explicitly name the product here.

Response: Thank you for pointing this out. We agree that the original wording referring to "satellite observations" was too general. In the revised Abstract, we now explicitly specify the satellite products used at different stages of the analysis. The relevant revised sentences are: (..., *yielding vertical profiles that are more consistent with MISR and CALIPSO observations.*).

18• **Lines 38–41:** It seems inconsistent that the study's final goal is to generate 3-D fire emissions, yet this section focuses only on analyzing drivers of SPH.

Response: We appreciate this comment and agree that the original wording did not sufficiently emphasize the role of SPH analysis in achieving the final objective of constructing a 3D emission inventory. In the revised Abstract, we clarify that SPH prediction and interpretation are not standalone objectives, but are explicitly used to constrain the vertical allocation of emissions. This is reflected in the following revised sentence: (*The fused FRP, together with ERA5 meteorology, drives a random forest (RF) model trained on MISR smoke plume heights (SPH) observations to predict SPH, which are then used to guide a multi-layer vertical allocation of emissions to construct the 3D emission inventory.; Compared with the widely used IS4FIRES v2.0 inventory, the resulting 3D SEAF dataset effectively mitigates near-surface–biased emission allocation and improves the representation of elevated smoke injection during peak burning periods, yielding vertical profiles that are more consistent with MISR and CALIPSO observations.*).

19• **Line 41:** The phrase "are anticipated to" is not appropriate, since you have already generated an observation-driven, hourly 3-D biomass-burning emission dataset for SEA.

Response: We thank the reviewer for this helpful comment. We agree that the phrase "are anticipated to" was overly tentative and did not accurately reflect the completed nature of the SEAF dataset. In the revised Abstract, this wording has been replaced with a definitive statement. The revised sentence now reads: (*By jointly mitigating systematic underestimation during key burning periods and alleviating low-altitude allocation bias while preserving elevated smoke occurrences, the SEAF inventory provides an observation-driven hourly 3 km 3D BB emission dataset for SEA, with improved temporal and vertical realism, supporting air quality and climate assessment applications.*)

**20·Lines 47–51:** If the manuscript focuses on validating CO, why not center the discussion on CO emissions in this section?

Response: Thank you for this suggestion. We agree that the role of CO should be made explicit in the Introduction. To address this point without expanding the scope of the section, we have revised the Introduction to explicitly include carbon monoxide (CO) among the major trace gases emitted from BB and to note that CO is commonly used as a tracer for BB (*Black carbon (BC) and primary organic aerosols (POA) derived from BB account for approximately 40% and 65% of global BC and POA emissions, respectively, while non-methane organic gases, carbon monoxide (CO, commonly used as a tracer for BB), and greenhouse gases such as methane (CH4), carbon dioxide (CO2), and nitrous oxide (N2O) contribute significantly to atmospheric chemistry and radiative forcing (Bond et al., 2013; Gkatzelis et al., 2024; Griffin et al., 2024).*). This minor revision clarifies the motivation for using CO as an observation-based tracer in the emission evaluation presented later in the manuscript.

**21• Lines 56–58:** The justification for selecting the SEA study area is not strong or straightforward.

Response: Thank you for this comment. We have revised the Introduction to clarify and strengthen the rationale for selecting SEA as the study region (**see our response to 2. Insufficient background and regional context and 9• Lines 321–324 — Transferability of methods**).

**22• Lines 56–81:** The background research for Asia is insufficient. Much prior work outside Asia is discussed, yet several important regional studies are missing.

Response: Thank you for this comment. We agree that the background research for Asia was insufficiently represented in the original version. In the revised Introduction, we have substantially expanded and restructured the regional background to better reflect prior work focused on SEA (**see our response to 2. Insufficient background and regional context and 9• Lines 321–324 — Transferability of methods**).

**23• Line 95:** Please add "three-dimensional" here for accuracy.

Response: Thank you for this helpful suggestion. We have added "3D" at Line 144 to improve accuracy.

**24• Line 97:** VIIRS and NOAA-20 should not be presented as parallel entities; one is a sensor, the other is a satellite platform.

Response: Thank you for pointing this out. We have revised the text (*SEAF was generated by fusing FRP data from the Advanced Himawari Imager (AHI), the VIIRS instruments onboard both the Suomi-NPP and NOAA-20 satellites.*).

**25• Lines 102–103:** Please remove the final sentence—its style is more appropriate for a proposal rather than a manuscript.

Response: Thank you for this suggestion. The final sentence has been removed.

**26• Lines 129–130:** The description of Himawari-8's spatial and temporal resolution is unnecessary here, as these specifics have already been provided.

Response: Thank you for the comment. We have revised the sentence to focus on the role of Himawari-8 observations over the SEA study region and removed redundant descriptions.

**27• Lines 131–132:** MODIS can detect nighttime events due to its ~1:30 a.m. overpass, although it may not capture all nighttime fires. This statement should be corrected.

Response: Thank you for this clarification. We have corrected the statement accordingly.

**28• Lines 163–165:** Please clarify what constitutes the testing dataset.

Response: Thank you for this comment. We have clarified the construction of the testing dataset in the revised manuscript by explicitly describing the quality control procedure, the train–test split (80% training and 20% independent testing), and the use of cross-validation and test-set performance metrics.

**29• Lines 171–174:** Why not directly include the common fire weather index variables (temperature, relative humidity, wind, and precipitation)?

Response: Thank you for this comment. We clarify that the ERA5 predictors used in this study already include near surface temperature, wind components, and precipitation. For atmospheric moisture, we used 2 m dew point temperature rather than relative humidity, since relative humidity can be directly derived from temperature and dew point and provides largely redundant information. This choice also helps reduce multicollinearity among predictors while retaining the physically relevant moisture constraint for plume development.

**30• Lines 177–180:** Why are widely used GFED products not included?

Response: Thank you for this constructive suggestion. **see our response to 7• Lack of GFED products in evaluation**.

• **Section 2.3.1:** If this section refers to VIIRS FRP data calibration, please revise the title to reflect that.

Response: Thank you for this suggestion. We have revised the section title to explicitly reflect that this section describes the cloud correction of VIIRS FRP data.

• **Line 5.7:** Please indicate the source of the test data.

Response: Thank you for pointing this out. We have clarified the source and partitioning of the test data in the revised manuscript. The test data is derived from the 2,127 MISR smoke plume height samples described in Section 2.1.3. To ensure a robust evaluation, the total dataset was randomly divided into a training set (80%) and an independent testing set (20%).

• **Figure 2:** Please add a legend explaining the colors.

Response: Thank you for this suggestion. We have revised the caption of Figure 2 to include a clear explanation of the color-coded framework. Specifically, we have added a description clarifying that:

- Green boxes represent input datasets/reference frameworks.
- Blue boxes denote intermediate data processing, calibration, and calculation steps.
- The pink circle represents the machine learning core (RF-SHAP framework).
- Orange boxes represent the final output products, including the predicted SPH and 3D emission inventory.

• **Figure 4:** Use either "correction" or "calibration" consistently throughout the text and figures.

Response: Thank you for pointing this out. We agree that the terminology should be used consistently and precisely. In this study, two different adjustment processes are applied to different sensors. Specifically, **VIIRS FRP is subject to cloud correction**, which aims to compensate for missing detections caused by cloud coverage without altering sensor-related biases. In contrast, Himawari FRP is adjusted through **cross-sensor calibration**, in which cloud-corrected VIIRS FRP serves as an external reference to correct systematic bias in Himawari observations. To avoid ambiguity, we have revised the figure caption and relevant text.

**Reference**

VIIRS I-Band 375 m Active Fire Data | NASA Earthdata: https://www.earthdata.nasa.gov/data/instruments/viirs/viirs-i-band-375-m-active-fire-data, last access: 22 December 2025.

Huang, H., Jin, Y., Sun, W., Gao, Y., Sun, P., and Ding, W.: Biomass burning in northeast China over two decades: Temporal trends and geographic patterns, Remote Sens., 16, 1911, https://doi.org/10.3390/rs16111911, 2024.

Schroeder, W., Oliva, P., Giglio, L., and Csiszar, I. A.: The new VIIRS 375m active fire detection data product: algorithm description and initial assessment, Remote Sens. Environ., 143, 85–96, https://doi.org/10.1016/j.rse.2013.12.008, 2014.

Schroeder, W., Giglio, L., Csiszar, I., and Tsidulko, M.: Algorithm theoretical basis document for NOAA NDE VIIRS I-band (375m) active fire, National Oceanic and Atmospheric Administration: Washington, DC, USA, 2020.

Vadrevu, K., Eaturu, A., Casadaban, E., Lasko, K., Schroeder, W., Biswas, S., Giglio, L., and Justice, C.: Spatial variations in vegetation fires and emissions in South and Southeast Asia during COVID-19 and pre-pandemic, Sci Rep, 12, 18233, https://doi.org/10.1038/s41598-022-22834-5, 2022.

Yin, S.: Biomass burning spatiotemporal variations over South and Southeast Asia, Environment International, 145, 106153, https://doi.org/10.1016/j.envint.2020.106153, 2020.

Zhang, T., Wooster, M. J., and Xu, W.: Approaches for synergistically exploiting VIIRS I- and M-Band data in regional active fire detection and FRP assessment: A demonstration with respect to agricultural residue burning in Eastern China, Remote Sensing of Environment, 198, 407–424, https://doi.org/10.1016/j.rse.2017.06.028, 2017.